# Algorithmic Guarantees for Distilling Supervised and Offline RL Datasets

**Aaryan Gupta**
Google DeepMind India
`aaryangupta@google.com`

**Rishi Saket**
Google DeepMind India
`rishisaket@google.com`

**Aravindan Raghuveer**
Google DeepMind India
`araghuveer@google.com`

## ABSTRACT

Given a training dataset, the goal of dataset distillation is to derive a synthetic dataset such that models trained on the latter perform as well as those trained on the training dataset. In this work, we develop and analyze an efficient dataset distillation algorithm for supervised learning, specifically regression in $\mathbb{R}^d$, based on matching the losses on the training and synthetic datasets with respect to a fixed set of randomly sampled regressors without any model training. Our first key contribution is a novel performance guarantee proving that our algorithm needs only $\tilde{O}(d^2)$ sampled regressors to derive a synthetic dataset on which the MSE loss of any bounded linear model is approximately the same as its MSE loss on the given training data. In particular, the model optimized on the synthetic data has close to minimum loss on the training data, thus performing nearly as well as the model optimized on the latter. Complementing this, we also prove a matching lower bound of $\Omega(d^2)$ for the number of sampled regressors showing the tightness of our analysis.

Our second contribution is to extend our algorithm to offline RL dataset distillation by matching the Bellman loss, unlike previous works which used a behavioral cloning objective. This is the first such method which leverages both, the rewards and the next state information, available in offline RL datasets, without any policy model optimization. We show similar guarantees: our algorithm generates a synthetic dataset whose Bellman loss with respect to any linear action-value predictor is close to the latter's Bellman loss on the offline RL training dataset. Therefore, a policy associated with an action-value predictor optimized on the synthetic dataset performs nearly as well as that derived from the one optimized on the training data. We conduct extensive experiments to validate our theoretical guarantees and observe performance gains on real-world RL environments with offline training datasets and supervised regression datasets.

## 1 INTRODUCTION

Reinforcement learning (RL) which is increasingly used to train very large machine learning models has two training paradigms: online and offline. While in online RL a policy model is trained while interacting with the environment, offline RL trains a model on data points collected from multiple trajectories of interactions with the environment and external entities (e.g. humans, AI agents). In many applications online RL is either not possible, very expensive or not scalable due the requirement of a dedicated agent interacting with the environment for the duration of model training. A key benefit of offline RL is its ability to ingest large amounts of diverse training data (Fu et al., 2021), and consequently, offline RL has become popular for training large models for tasks in for e.g. natural language processing, computer vision, robotics etc. (Levine et al., 2020).

However, the use of large scale datasets presents challenges related to their storage, management as well the computational expense incurred in their use for training multiple models with different

hyperparameters and architectures. One way to mitigate this issue is to create a smaller synthetic dataset derived from the training dataset i.e., a *distillation* of the latter. Dataset distillation (DD) for supervised learning has been extensively studied in previous works (e.g. (Wang et al., 2018; Sachdeva & McAuley, 2023; Lei & Tao, 2024)) which have developed a variety of methods based on matching the loss gradients or feature-embeddings between the training and synthetic datasets by optimizing the latter. This is done either with respect to networks being trained simultaneously using bi-level optimization, or a fixed collection of sampled networks. Distinct from sampling, coreset or random projection methods, DD techniques generate synthetic data via optimization with respect to networks which may also be trained in the process. These DD techniques have been shown to perform well on real-world supervised datasets in terms of the quality as well as the size of the generated synthetic dataset as the latter is explicitly optimized.

While some works (e.g. (Nguyen et al., 2021; Chen et al., 2024)) have also provided theoretical performance guarantees, most DD techniques are heuristics albeit with empirically observed gains.

For offline RL, a few recent works have proposed methods using behavioral cloning (BC) for distilling an offline training dataset comprising trajectories of (state, action)-pairs. In (Lei et al., 2024), the authors propose optimizing the synthetic dataset using policy-based BC loss leveraging action-value weights learnt from offline RL. On the other hand, the method proposed by Light et al. (2024) extends the matching loss gradients approach to offline RL. In particular, it optimizes a synthetic dataset to match the BC loss gradients on the offline and synthetic data, where the gradients are with respect to the parameters of sampled action predictor networks.

As can be seen, the state of research into offline RL dataset distillation, while nascent, is also unsatisfying. Firstly, the BC does not leverage the observed reward that is usually available in offline RL datasets, and typically only perform well when the training dataset is generated by expert policies. Secondly, there is a sparsity of theoretical performance guarantees for the DD techniques for offline RL as well as supervised learning DD. For linear regression, the work of Chen et al. (2024) proves efficiency guarantees assuming however that a trained model is available, while the work of Nguyen et al. (2021) – while not training a model – proves convergence only of the synthetic feature-vectors for given synthetic labels. For offline RL, the work of Lei et al. (2024) provides analytical guarantees, while requiring an trained policy model on the given training dataset. To the best of our knowledge, there are no provably efficient DD algorithms known without model training for either supervised learning or offline RL.

**Our Contributions.** In this work we make progress towards bridging the above gaps in our understanding through the following contributions:

**Dataset Distillation in supervised regression.** We propose an algorithm minimizing the squared difference of MSE losses between the training and the synthetic datasets with respect to a fixed set of randomly sampled models. While our algorithm is along the line of previous DD approaches using loss, gradient loss or embedding matching techniques, our contribution is to prove that it admits efficiency and performance guarantees for linear regression. In particular, we prove that the optimization objective is convex and tractable, and in $d$-dimensions optimizing the synthetic dataset w.r.t. $\tilde{O}(d^2)$[1] randomly sampled regressors suffices to guarantee that any bounded linear regressor has approximately the same MSE loss on the training and synthetic datasets. Therefore, the optimum linear regressor on the synthetic dataset is close to that on the training dataset. This is the first performance and efficiency guarantee for supervised DD without model training.
We also prove a lower bound of $\Omega(d^2)$ on the number of randomly sampled linear regressors to obtain a good quality synthetic dataset. This result shows the tightness of our algorithmic guarantees.

**Dataset Distillation in offline RL.** We extend the above approach for dataset distillation in supervised regression to offline RL DD by matching the Bellman loss on the training and synthetic datasets w.r.t. a collection of randomly sampled linear $Q$-value predictors over $\mathbb{R}^d$. The performance guarantees are in a similar vein as the supervised regression case, though more involved due to the $\max$ term in the Bellman Loss formulation.

Nevertheless, we obtain a performance guarantee showing that minimizing to a small value the difference of the Bellman Losses with respect to $\exp(O(d \log d))$ randomly sampled $Q$-value predictors (without model training) generates a synthetic dataset whose Bellman Loss w.r.t. any linear bounded

---

[1] $\tilde{O}$ hides polylogarithmic factors

$Q$-value predictor is a close approximation to that on the training dataset. Consequently, the Bellman Loss minimizer $Q$-value predictor on the synthetic dataset is close to optimal on the training dataset.

**Offline RL with decomposable feature-maps.** We consider a natural setting where the state-action feature-map is the sum of individual feature-maps of the state and action. In this scenario, we prove that our method requires only $\tilde{O}(d^2)$ sampled linear $Q$-value predictions, instead of the exponentially many required in the general case. Additionally, we show that the finding the synthetic dataset is a tractable convex optimization when the state and action feature-maps are linear.

**Empirical Evaluations.** We conduct experiments on supervised regression and offline RL datasets to validate our proposed algorithms, showing that our techniques obtain improved performance over standard baselines, both in terms of quality and size of the generated datasets. Notably, our experiments demonstrate the effectiveness of our algorithms using very few sampled models in both supervised regression and offline RL settings. Although the guarantees are proven for linear models, we show that our algorithms work well in practice with non-linear neural networks.

## 2 PREVIOUS RELATED WORK

**Dataset Distillation.** The work of Wang et al. (2018) introduced the problem of *distilling* a dataset into a representative (and smaller) synthetic dataset, in the setting of supervised learning. This and other works e.g. (Deng & Russakovsky, 2022) use a bi-level optimization formulation in which the model is optimized on the training dataset while the synthetic dataset is optimized on the trained model. A related set of methods rely on matching various properties of the synthetic dataset with the training dataset. In particular, the work of Zhao et al. (2021); Zhao & Bilen (2021) matches the model's loss gradients on the training and synthetic datasets as an optimization over the latter, while the model is alternately optimized over the training dataset. In a similar vein, the work of Wang et al. (2022) aligns the feature distribution of the two datasets in a dynamic bi-level optimization approach, while the works of Cazenavette et al. (2022); Cui et al. (2023) match the training trajectory of an initial model optimized on the training dataset with its training trajectory on the synthetic dataset, by optimizing on the latter. Unlike the mentioned works optimizing a model along with the synthetic dataset, the work of Zhao & Bilen (2023) instead matched the feature-distributions on the training and synthetic datasets with respect to a fixed collection of randomly sampled networks. For linear ridge regression, the work of Nguyen et al. (2021) implicitly matched the regression losses by minimizing a surrogate objective, while proving convergence of synthetic feature-vectors given the synthetic labels. More recently, Chen et al. (2024) analytically gave an efficient synthetic dataset generation algorithm for linear ridge regression, requiring however access to the optimal regression model for the training dataset.

Closely related to DD for linear regression is *matrix sketching* which provides a principled way to reduce the dimensionality (or size) of training data. By applying randomized projections (e.g., the Johnson–Lindenstrauss transform (Johnson & Lindenstrauss, 1984; Sarlós, 2006)) or leverage-score sampling (Drineas et al., 2006; Mahoney, 2011), one can construct a much smaller *sketch* of the original data while provably preserving the solution quality of least-squares regression.

However, these guarantees are largely limited to linear and convex models. In contrast, DD methods can be applied to neural networks, albeit without comparable theoretical guarantees. Additionally, DD techniques optimize the synthetic dataset, and have been shown to to work well in practice on real data in terms of size compression and quality preservation.

**Offline RL.** A key advantage is that offline RL is adaptable to data generated by sub-optimal policies (Levine et al., 2020; Fu et al., 2021), while also being scalable to large datasets (Lange et al., 2012). On the other hand, the lack of online exploration can lead to less generalizable policies being trained due to distributional shifts. To address this, several techniques based on constraining the policy (Fujimoto & Gu, 2021; Tarasov et al., 2023) or regularization of $Q$-value predictors have been developed (Kumar et al., 2020; Kostrikov et al., 2022). A major practical consideration is the exponential growth in dataset sizes, along with which the associated challenges of storage, transfer and training on datasets as well as maintaining privacy controls have only multiplied. Dataset distillation has been proposed to tackle these problems, with existing works developing behavioral cloning methods for distilling an offline training dataset comprising trajectories of (state, action)-pairs. While Lupu et al. (2024) propose behavior distillation for online RL, the work of Lei et al. (2024)

tackles the offline RL case by first training an action-value predictor and using action-value weighted BC to optimize the synthetic dataset. On the other hand, the method proposed by Light et al. (2024) extends the matching loss gradients approach to offline RL. In particular, it optimizes a synthetic dataset to match the behavioral cloning loss gradients on the offline and synthetic data, where the gradients are with respect to the parameters of sampled action predictor networks.

## 3 PROBLEM DEFINITION

We will use $\mathcal{B}(t, r) := \{\mathbf{x} \in \mathbb{R}^t \mid \|\mathbf{x}\|_2 \leq r\}$ to denote the $\ell_2$-ball in $\mathbb{R}^t$ centered at $\mathbf{0}$ of radius $r$.

**Supervised Learning Dataset Distillation.** We consider regression tasks over $d$-dimensional real-vectors and real-valued labels. For an $n$-point dataset $D \in (\mathbb{R}^d \times \mathbb{R})^n$, and a predictor $f : \mathbb{R}^d \to \mathbb{R}$, the mean squared error (MSE) of $f$ on $D$ is $L_{\mathrm{mse}}(D, f) := (1/n) \sum_{(\mathbf{x}, y) \in D} (y - f(\mathbf{x}))^2$. We assume that the training datapoints are norm bounded by $B$ and labels have magnitude at most $b$ for some parameters $B$ and $b$, and impose the same restriction on the synthetic dataset. Let $D_{\mathrm{train}}^{\mathsf{sup}} = \{(\mathbf{x}_i, y_i) \in \mathcal{B}(d, B) \times [-b, b]\}_{i=1}^n$ be an $n$-sized training dataset, while we denote an $m$-sized synthetic dataset by $D_{\mathrm{syn}}^{\mathsf{sup}} = \{(\mathbf{z}_i, \hat{y}_i) \in \mathcal{B}(d, B) \times [-b, b]\}_{i=1}^m$. By appending a 1-valued coordinate to feature-vectors we can omit the constant offset and restrict ourselves to linear regressors of the form $\mathbf{v}^\top \mathbf{x}$ as a prediction for the label $y$. We impose $\|\mathbf{v}\|_2 \leq 1$ as a bound on the norm, and let $\mathcal{F}_0$ denote the class of such regressors. We define the *supervised regression dataset distillation* problem as: for a parameter $\varepsilon$, given $D_{\mathrm{train}}^{\mathsf{sup}}$, compute $D_{\mathrm{syn}}^{\mathsf{sup}}$ such that

$$L_{\mathrm{mse}}^{\mathrm{err}}(D_{\mathrm{train}}^{\mathsf{sup}}, D_{\mathrm{syn}}^{\mathsf{sup}}, f) := \left(L_{\mathrm{mse}}(D_{\mathrm{train}}^{\mathsf{sup}}, f) - L_{\mathrm{mse}}(D_{\mathrm{syn}}^{\mathsf{sup}}, f)\right)^2 \leq \varepsilon, \qquad \text{for all } f \in \mathcal{F}_0. \quad (1)$$

**Offline RL Dataset Distillation.** Consider a Markov Decision Process (MDP) given by $\langle \mathcal{S}, \mathcal{A}, \mathcal{P}, \mathcal{R}, \gamma \rangle$, where (i) $\mathcal{S}$ is the set of states, (ii) $\mathcal{A}$ is the set of actions, (iii) $\mathcal{P}$ is the transition probability i.e., $\mathcal{P}(s' \mid s, a)$ denotes the probability of transitioning to state $s'$ from state $s$ on action $a$, (iv) $\mathcal{R}$ is the reward function, where $\mathcal{R}(s, a) \in [0, R_{\max}]$ is the reward obtained on action $a$ at state $s$, and (v) $\gamma$ is the discount factor. A *policy* $\pi$ is a mapping from states to actions, and the goal is to maximize at each state its *value function*: $v_\pi(s) = \mathbb{E}_\pi \left[\sum_{t=0}^\infty \gamma^t R_t \mid s_0 = s\right]$ where $R_t$ is the reward at step $t$ under the policy starting from state $s$. The action-value function is the expected sum of discounted rewards starting from a state and a specific action i.e., $q_\pi(s, a) = \mathbb{E}_\pi \left[\sum_{t=0}^\infty \gamma^t R_t \mid s_0 = s, a_0 = a\right]$. On the other hand, given an action value function predictor $f : \mathcal{S} \times \mathcal{A} \to \mathbb{R}$, the corresponding greedy policy $\pi'$ given by $\pi'(s) \in \mathrm{argmax}_a f(s, a)$ always yields at least as much expected discounted reward starting from any state as $\pi$. In offline RL, a dataset $D^{\mathsf{orl}}$ consists of a collection of state, action, reward and next state tuples of the form $(s, a, r, s')$, with generated by some (non-optimal) policies. The goal is to learn a policy from this dataset which maximizes the value function. We cast this problem as deriving a action-value predictor from $D^{\mathsf{orl}}$, and taking the greedy policy with respect to it. Under reasonable assumptions on the MDP and $D^{\mathsf{orl}}$, the performance of an action-value predictor $f$ is measured by the Bellman loss (see Appendix G.1 for details) given by:

$$L_{\mathrm{Bell}}(D^{\mathsf{orl}}, f) = \mathbb{E}_{(s, a, r, s') \leftarrow D^{\mathsf{orl}}} \left[\left(f(s, a) - r - \gamma \max_{a' \in \mathcal{A}} f(s', a')\right)^2\right]. \quad (2)$$

*Feature Map and linear action-value predictors.* We assume that $\mathcal{S}, \mathcal{A} \subseteq \mathcal{B}(d_0, B_0)$, and that $\phi : \mathcal{B}(d_0, B_0)^2 \to \mathbb{R}^d$ is a given feature-map s.t. $\|\phi(s, a)\|_2 \leq B$ for any $(s, a)$ in its domain, for some parameters $B_0, d_0$ and $B$. An action-value predictor $f$ is given by $f(s, a) := \mathbf{v}^\top \phi(s, a)$, for $(s, a) \in \mathcal{B}(d_0, B_0)^2$, and we restrict ourselves to the class of such predictors $\mathcal{Q}_0$ satisfying $\|\mathbf{v}\|_2 \leq 1$.
*Datasets.* The training dataset is $D_{\mathrm{train}}^{\mathsf{orl}} = \{(s_i, a_i, r_i, s_i')\}_{i=1}^n \subseteq \mathcal{S} \times \mathcal{A} \times [0, R_{\max}] \times \mathcal{S}$ of state, action, reward and next state tuples. Since $\mathcal{S}$ or $\mathcal{A}$ could either be discrete or non-convex, for tractable optimization the synthetic dataset $D_{\mathrm{syn}}^{\mathsf{orl}}$ is allowed to consist of tuples $\{(\hat{s}_i, \hat{a}_i, \hat{r}_i, \hat{s}_i') \in \mathcal{B}(d_0, B_0) \times \mathcal{B}(d_0, B_0) \times [0, R_{\max}] \times \mathcal{B}(d_0, B_0)\}$. With this, $L_{\mathrm{Bell}}$ is can defined for $D_{\mathrm{syn}}^{\mathsf{orl}}$ and any $f \in \mathcal{Q}_0$ as:

$$L_{\mathrm{Bell}}(D_{\mathrm{syn}}^{\mathsf{orl}}, f) = \mathbb{E}_{(\hat{s}, \hat{a}, \hat{r}, \hat{s}') \leftarrow D_{\mathrm{syn}}^{\mathsf{orl}}} \left[\left(f(\hat{s}, \hat{a}) - \hat{r} - \gamma \max_{\hat{a}' \in \mathcal{A}} f(\hat{s}', \hat{a}')\right)^2\right]. \quad (3)$$

Note that in the above, the maximization inside the loss is taken over the original set of actions, consistent with the definition of the Bellman loss. With the above setup, we define the *offline RL*

*dataset distillation problem* as follows: For a parameter $\varepsilon$, given $D_{\text{train}}^{\text{orl}}$, compute $D_{\text{syn}}^{\text{orl}}$ such that

$$L_{\text{Bell}}^{\text{err}}(D_{\text{train}}^{\text{orl}}, D_{\text{syn}}^{\text{orl}}, f) := \left(L_{\text{Bell}}(D_{\text{train}}^{\text{orl}}, f) - L_{\text{Bell}}(D_{\text{syn}}^{\text{orl}}, f)\right)^2 \leq \varepsilon, \qquad \text{for all } f \in \mathcal{Q}_0. \tag{4}$$

## 4 OUR RESULTS

**Supervised Learning Dataset Distillation.** For convenience, we shall employ a homogeneous formulation i.e., with 0 label, using the concatenation $\zeta : \mathbb{R}^d \times \mathbb{R} \to \mathbb{R}^{d+1}$ given by $\zeta(\mathbf{x}, y) = (x_1, \ldots, x_d, y)$ where $\mathbf{x} = (x_1, \ldots, x_d)$. Observe that $\mathbf{r}^\mathsf{T}\zeta(\mathbf{x}, y) = \mathbf{v}^\mathsf{T}\mathbf{x} - y = f(\mathbf{x}) - y$ where $\mathbf{r} = (v_1, \ldots, v_d, -1)$, and $f \in \mathcal{F}_0$ s.t. $f(\mathbf{x}) = \mathbf{v}^\mathsf{T}\mathbf{x}$. Further, if $\|\mathbf{v}\|_2 \leq 1$ then $1 \leq \|\mathbf{r}\|_2 \leq 2$. Let $\mathcal{F}$ be the class regressors with target label 0 where each $h \in \mathcal{F}$ is given by $h(\mathbf{x}, y) := \mathbf{r}^\mathsf{T}\zeta(\mathbf{x}, y)$ where $1 \leq \|\mathbf{r}\|_2 \leq 2$. One can thus extend the notion of the MSE loss $L_{\text{mse}}$ to $\mathcal{F}$ by letting $L_{\text{mse}}(D, h) := \mathbb{E}_{(\mathbf{x},y) \in D}\left[h(\mathbf{x}, y)^2\right]$. We also define $G$ to be a distribution over regressors where a random $g \in G$ is given by sampling $\mathbf{r} \sim N(0, 1/(d+1))^{d+1}$ u.a.r. and letting $g(\mathbf{x}, y) := \mathbf{r}^\mathsf{T}\zeta(\mathbf{x}, y)$. With this setup, we prove the following theorem.

**Theorem 4.1.** *Let $D_{\text{train}}^{\text{sup}}$ the training dataset as described above. For any $\Delta > 0$ and $\delta > 0$, let $g_1, \ldots, g_k$ be iid regressors sampled from $G$ for some $k = O\left(d^2 \log(d(B + b)/\Delta) \log(1/\delta)\right)$. Then, with probability $1 - \delta$ over the choice of $g_1, \ldots, g_k$, if there exists $D_{\text{syn}}^{\text{sup}}$ s.t.*

$$\sum_{j=1}^{k} L_{\text{mse}}^{\text{err}}(D_{\text{train}}^{\text{sup}}, D_{\text{syn}}^{\text{sup}}, g_j) \leq \Delta', \tag{5}$$

*then, for all $h \in \mathcal{F}$, $L_{\text{mse}}^{\text{err}}(D_{\text{train}}^{\text{sup}}, D_{\text{syn}}^{\text{sup}}, h) \leq \Delta$, where $\Delta' = \Delta k/O(d^2)$, in particular this holds also for all $f \in \mathcal{F}_0$.*

It can be seen that $L_{\text{mse}}^{\text{err}}(D_{\text{train}}^{\text{sup}}, D_{\text{syn}}^{\text{sup}}, g)$ is a convex function over the points of $D_{\text{syn}}^{\text{sup}}$ (see Appendix G.2 for an explanation) and therefore the LHS of equation 5 is also convex and can be minimized efficiently. Based on this we provide the corresponding Algorithm 1.

---

**Algorithm 1:** Supervised Regression Dataset Distillation

---

**Input:** $d, k, m \in \mathbb{Z}^+, D_{\text{train}}^{\text{sup}} \in (\mathcal{B}(d, B) \times [-b, b])^n$
1. Sample iid at random $g_1, \ldots, g_k$ from $G$.
2. Output $\quad \text{argmin}_{D_{\text{syn}}^{\text{sup}} \in (\mathcal{B}(d,B) \times [-b,b])^m} \sum_{j=1}^{k} L_{\text{mse}}^{\text{err}}(D_{\text{train}}^{\text{sup}}, D_{\text{syn}}^{\text{sup}}, g_j)$.

---

**Lower Bound.** Complementing the above algorithmic result, we prove the following matching (up to logarithmic factors) lower bound on the number of sampled regressors.

**Theorem 4.2.** *For any positive integer $d$, there exists $D_{\text{train}}^{\text{sup}}$ of $q = (d + 1)$ points of the form $\mathbf{z} = (\mathbf{x}, y) \in \mathbb{R}^q$ where $\mathbf{x} \in \mathbb{R}^d, y \in \mathbb{R}$, each of Euclidean norm $\|\mathbf{z}\|_2 \leq 2$ such that for any choice of homogeneous (i.e., target label 0) regressors $\{f_t : \mathbb{R}^q \to \mathbb{R} \mid f_t(\mathbf{z}) := \mathbf{v}_t^\mathsf{T}\mathbf{z}, \text{ where } \mathbf{v}_t \in \mathbb{R}^q\}_{t=1}^{T}$, where $T < q(q + 1)/2$, there exists:*

- *$D_{\text{syn}}^{\text{sup}}$ of $q$ points in $\mathbb{R}^q$ each of Euclidean norm $\leq 2$, and*
- *a regressor $f_0 : \mathbb{R}^q \to \mathbb{R}$ given by $f_0(\mathbf{z}) := \mathbf{v}_0^\mathsf{T}\mathbf{z}$ for a unit vector $\mathbf{v}_0 \in \mathbb{R}^q$,*

*satisfying*

$$L_{\text{mse}}^{\text{err}}(D_{\text{train}}^{\text{sup}}, D_{\text{syn}}^{\text{sup}}, f_t) = 0, \ \forall t \in \{1, \ldots, T\} \qquad \text{and} \qquad L_{\text{mse}}^{\text{err}}(D_{\text{train}}^{\text{sup}}, D_{\text{syn}}^{\text{sup}}, f_0) \geq 1/(4q^2). \tag{6}$$

Informally, the above theorem constructs a training dataset such that for any set of less than $q(q+1)/2$ linear regressors one can choose a synthetic dataset on which each of the linear regressors have the same loss as on the training dataset, while there exists a regressor that has significantly different losses. This implies a lower bound of $\Omega(d^2)$ for $k$ in Theorem 4.1.

Note that Theorem 4.1 states that as long as there exists *any* $D_{\text{syn}}^{\text{sup}}$ of size $m$ such that equation 5 is satisfied, the implications of the theorem hold i.e. $D_{\text{syn}}^{\text{sup}}$ is a good distilled synthetic dataset. There is no restriction on $m$ as long as equation 5 is satisfied and Theorem 4.1 does not explicitly impose any

lower bound on $m$. However, depending on $D_{\text{train}}^{\text{sup}}$, it may not always be possible to satisfy equation 5 for small values of $m$. Specifically, we show in Appendix D a lower bound on based on the spectral properties of the gram matrix of $D_{\text{train}}^{\text{sup}}$. In the worst case, we show a lower bound of $m \geq d + 1$ and show that it is possible to achieve this lower bound implying its tightness.

**Offline RL Dataset Distillation.** For convenience, we define the following modified Bellman loss which incorporates a scale factor $\lambda \in \mathbb{R}$ for the reward in the usual Bellman loss:

$$\hat{L}_{\text{Bell}}(D, f, \lambda) := \mathbb{E}_{(s,a,r,s') \leftarrow D} \left[ \left( f(s,a) - \lambda r - \gamma \max_{a' \in \mathcal{A}} f(s',a') \right)^2 \right]. \tag{7}$$

To state our result, we need the *pseudo-dimension* Pdim (see Appendix A.1) of the above loss restricted to single points. In particular, $\mathcal{U}$ be class of mappings $u : \mathcal{B}(d_0, B_0) \times \mathcal{B}(d_0, B_0) \times [0, R_{\max}] \times \mathcal{B}(d_0, B_0) \to \mathbb{R}$ where each $u \in \mathcal{U}$ is defined by $u(s,a,r,s') := \hat{L}_{\text{Bell}}(\{(s,a,r,s')\}, f, \lambda)$ for some $f \in \mathcal{Q}_0$ and $\lambda \in [-1, 1]$. Let $p := \text{Pdim}(\mathcal{U})$. Further, we assume that $\phi$ is $L$-Lipschitz. We also define:

$$\hat{L}_{\text{Bell}}^{\text{err}}(D_{\text{train}}^{\text{orl}}, D_{\text{syn}}^{\text{orl}}, f, \lambda) = \left( \hat{L}_{\text{Bell}}(D_{\text{train}}^{\text{orl}}, f, \lambda) - \hat{L}_{\text{Bell}}(D_{\text{syn}}^{\text{orl}}, f, \lambda) \right)^2 \tag{8}$$

We define a distribution $\tilde{H}(\sigma)$ over (predictor, scalar) pairs in which a random $(f, \lambda)$ is sampled by independently choosing $\mathbf{r} \sim N(0, \sigma^2)^d$ and $\lambda \sim N(0, \sigma^2)$ and defining $f(s,a) := \mathbf{r}^\mathsf{T} \phi(s,a)$. For simplicity we define $H := \tilde{H}(1)$ Using this, we have the following theorem.

**Theorem 4.3.** *Let $D_{\text{train}}^{\text{orl}}$ the offline RL training dataset as described above. For any $\Delta > 0$ and $\delta > 0$, let $(f_1, \lambda_1), \ldots, (f_k, \lambda_k)$ be iid samples from $H$ for some $k = (1/\nu)^{O(d \log d)} O\left(d_0 p \log(1/\nu) + d \log(B_0 L(B + R_{\max})/\Delta)\right) \log(1/\delta)$, where $\nu := \Delta/(B + R_{\max})^4$. Then, with probability $1 - \delta$, if there exists $D_{\text{syn}}^{\text{orl}}$ s.t.*

$$\sum_{j=1}^{k} \hat{L}_{\text{Bell}}^{\text{err}}(D_{\text{train}}^{\text{orl}}, D_{\text{syn}}^{\text{orl}}, f_j, \lambda_j) \leq \Delta' \tag{9}$$

*then, for all $f \in \mathcal{Q}_0$, $L_{\text{Bell}}^{\text{err}}(D_{\text{train}}^{\text{orl}}, D_{\text{syn}}^{\text{orl}}, f) \leq \Delta$, where $\Delta' = (\Delta/O(1))$.*

The following is the distillation algorithm whose guarantees follow directly from Theorem 4.3.

---

**Algorithm 2:** Offline RL Dataset Distillation

---

**Input:** $d, k, m \in \mathbb{Z}^+$, feature-map $\phi$, $D_{\text{train}}^{\text{orl}} \in (\mathcal{S} \times \mathcal{A} \times [0, R_{\max}] \times \mathcal{S})^n$.
1. Sample iid at random $(f_1, \lambda_1) \ldots, (f_k, \lambda_k)$ from $H$.
2. Output $\quad \text{argmin}_{D_{\text{syn}}^{\text{orl}} \in (\mathcal{B}(d_0, B_0) \times \mathcal{B}(d_0, B_0) \times [0, R_{\max}] \times \mathcal{B}(d_0, B_0))^m} \sum_{j=1}^{k} \hat{L}_{\text{Bell}}^{\text{err}}(D_{\text{train}}^{\text{orl}}, D_{\text{syn}}^{\text{orl}}, f_j, \lambda_j)$

---

**Offline RL with decomposable feature-map.** We consider the natural case when $\phi$ is decomposable i.e., $\phi(s,a) := \phi_1(s) + \phi_2(a) \in \mathbb{R}^d$ for all $s \in \mathcal{S}, a \in \mathcal{A}$ for some mappings $\phi_1, \phi_2 : \mathcal{B}(d_0, B_0) \to \mathbb{R}^d$. Further, we say that $\phi$ is linear and decomposable if $\phi_1$ and $\phi_2$ are linear maps.

**Theorem 4.4.** *Consider the case when $\phi$ is decomposable as defined above and let $D_{\text{train}}^{\text{orl}}$ the offline RL training dataset. For any $\Delta > 0$ and $\delta > 0$, let $(f_1, \lambda_1), \ldots, (f_k, \lambda_k)$ be iid samples from $\tilde{H}(1/\sqrt{d+1})$ for some $k = O\left(d^2 \log(d(B + R_{\max})/\Delta) \log(1/\delta)\right)$. Then, with probability $1 - \delta$, if there exists $D_{\text{syn}}^{\text{orl}}$ s.t.*

$$\sum_{j=1}^{k} \hat{L}_{\text{Bell}}^{\text{err}}(D_{\text{train}}^{\text{orl}}, D_{\text{syn}}^{\text{orl}}, f_j, \lambda_j) \leq \Delta' \tag{10}$$

*satisfying*

$$\mathbb{E}_{D_{\text{syn}}^{\text{orl}}}[(\phi_1(s), \phi_2(a), r, \phi_1(s'))] = \mathbb{E}_{D_{\text{train}}^{\text{orl}}}[(\phi_1(s), \phi_2(a), r, \phi_1(s'))] \tag{11}$$

*then, for all $f \in \mathcal{Q}_0$, $L_{\text{Bell}}^{\text{err}}(D_{\text{train}}^{\text{orl}}, D_{\text{syn}}^{\text{orl}}, f) \leq \Delta$, where $\Delta' = \Delta k/O(d^2)$.*

In particular, the above theorem shows that the optimization in Algorithm 2 constrained by equation 11, in the case of decomposable feature-map $\phi$, requires only $\tilde{O}(d^2)$ sampled action-value predictors. Further, when $\phi$ is linear and decomposable i.e., $\phi_1$ and $\phi_2$ are linear maps, then equation 11 is a linear constraint in the points $D_{\text{syn}}^{\text{orl}}$ and it can be shown (see Appendix F for details) that the optimization in Algorithm 2 constrained by equation 11 is convex.

**Discussion of Our Results.** Theorem 4.1 and Algorithm 1 together provide an efficient supervised dataset distillation algorithm (for linear regression) with performance guarantees. Specifically, we show that with high probability over $\tilde{O}(d^2)$ sampled regressors, optimizing a convex objective over $D_{\text{syn}}^{\text{sup}}$ to minimize the sum of $L_{\text{mse}}^{\text{err}}$ for the sampled regressors, is sufficient to obtain a high quality synthetic dataset. While our method adapts the model-training free approach of Zhao & Bilen (2023) to loss matching, our theoretical guarantees are qualitatively different from those of Nguyen et al. (2021) which showed convergence of synthetic features assuming the synthetic labels are given, and of Chen et al. (2024) who assumed the availability of the optimal trained model. We show that our analysis is tight (upto polylogarithmic factors) by proving a $\Omega(d^2)$ lower bound on the number of sampled regressors in Theorem 4.2. In addition, we also provide a lower bound of $O(d)$ on the size of the synthetic dataset needed (in Appendix D). We note that Theorem 4.1 shows that the randomness (or sample complexity) required is proportional to $\log(1/\Delta)$ where $\Delta$ is the error, which is quantitatively better than sample size proportional to $1/\Delta$ achieved by matrix sketching techniques (see Garg et al. (2024), Wang et al. (2017), Clarkson & Woodruff (2009)). Our theoretical analysis of the supervised regression relies on anti-concentration of $L_{\text{mse}}^{\text{err}}$ with respect to random linear regressors, and can be applied to any class of non-linear regressors that admit similar anti-concentration properties. Our theoretical analysis of the supervised regression setting can be carried out for any class of non-linear predictors that admit anti-concentration of $L_{\text{mse}}^{\text{err}}$ over the choice of the predictor's random weights. In the linear setting, $L_{\text{mse}}^{\text{err}}$ is a degree 4 polynomial in the weights of the linear predictor and we use the Carbery-Wright inequality for polynomials (Theorem A.3) to get anti-concentration in Lemma C.1. If some class of neural networks can be approximated by polynomials of bounded degree (as suggested by Frances et al. (2022)) in the weights of the neural network which can be sampled randomly, our techniques could be applied. However, proving such anti-concentration for general neural networks is beyond the scope of this paper and we defer this line of work to the future.

For offline RL, Theorem 4.3 and Algorithm 2 give the first dataset distillation algorithm with rigorous performance bounds, without model training. Our novel approach based on matching the Bellman loss w.r.t. sampled $Q$-value predictors differs from the previous behavioral cloning (BC) based methods of Light et al. (2024) which matches the loss gradients of the BC objective and of Lei et al. (2024) which optimizes the synthetic dataset using an action-value weighted BC loss requiring a trained action-value predictor.

However, the Bellman Loss involves a max term and thus cannot be well represented as a low degree polynomial in the weights of the predictor. Due to this, polynomial anti-concentration (used in the supervised regression case) cannot be applied and the proof is via a conditioning argument. In effect, our algorithm requires sampling $\exp(O(d \log d))$ predictors in the worst case where $d$ is the output dimension of the feature-map. We note that a brute force approach would be to choose a net over the predictors $\mathcal{Q}_0$ instead of sampling. However, that would necessarily require $\exp(d)$ such predictors, while in practice a smaller number of randomly sampled predictors suffices, as demonstrated in our experimental evaluations.

While the bound in Theorem 4.3 is indeed less efficient than $\tilde{O}(d^2)$ sample complexity of the supervised setting, in practice the features are mapped into a smaller dimension than the input mitigating this to some extent. Further, in Theorem 4.4, we prove similar $\tilde{O}(d^2)$ in the case of decomposable feature-map $\phi$, and show that the optimization is convex when $\phi$ is linear and decomposable. This is a natural assumption and is the default feature map in applied RL for most value based deep RL approaches. For example, the highly cited work of (Lillicrap et al., 2016) explicitly describes concatenating the action and state embeddings. A common trick used in practice and in our experiments is to one-hot encode the actions and concatenate them with the state embedding to obtain the feature map. Additive feature maps are also explicitly studied in (Zhang et al., 2020) and (Yang & Wang, 2019). Our analysis uses a conditioning argument which leverages the linearity of the predictor for using Gaussian anti-concentration. The anti-concentration is used to show that the sampled linear predictors lie in an appropriate subset with non-trivial probability. We believe Theorem 4.4 provides

evidence that proving appropriate Lipschitzness and anti-concentration of a non-linear predictor's output w.r.t. its weights, can be used to extend our arguments to such predictors, e.g. neural networks. We however, defer a more detailed study of the non-linear predictor case for the future.

**Organization of the Paper.** The proofs of Theorems 4.1, 4.2, 4.3 and 4.4 are included in Appendices C, D.2, E and F respectively. In the next section however, we provide an informal description of the proof techniques. A subset of the experimental evaluations are included in Section 6 while additional experiments and further details are deferred to Appendix H.

# 5 OVERVIEW OF OUR TECHNIQUES

**Proof Outline of Theorem 4.1.** The proof proceeds by contradiction: suppose there is some $h \in \mathcal{F}$ and $D_{\text{syn}}^{\text{sup}}$ s.t. $L_{\text{mse}}^{\text{err}}(D_{\text{train}}^{\text{sup}}, D_{\text{syn}}^{\text{sup}}, h) > \Delta$. Letting $g$ be a random sample from $G$, we show using algebraic manipulations and properties of the Gaussian distribution that $L_{\text{mse}}^{\text{err}}(D_{\text{train}}^{\text{sup}}, D_{\text{syn}}^{\text{sup}}, g)$ is a degree-4 polynomial $v^2$ in Gaussian variables, such that $\mathbb{E}[v^2] > \Delta/O(d^2)$. This can be used along with the Carbery-Wright anti-concentration bound for Gaussian polynomials to show that $\Pr[v^2 > \Delta/O(d^2)] \geq 1/3$. Since $\{g_j\}_{j=1}^k$ are iid samples from $G$, using Chernoff Bound, the probability that there exists $\Omega(k)$ many $j \in [k]$ satisfying $L_{\text{mse}}^{\text{err}}(D_{\text{train}}^{\text{sup}}, D_{\text{syn}}^{\text{sup}}, g_j) > \Delta/O(d^2)$ is at least $1 - \exp^{-\Omega(k)}$. To complete the argument, we require a union bound of this error probability over all $h \in \mathcal{F}$ and $D_{\text{syn}}^{\text{sup}}$, which we do by constructing finegrained nets as follows.
*Net over all $h \in \mathcal{F}$*: Since $h \in \mathcal{F}$ is given by some $\mathbf{r} \in \mathbb{R}^{d+1}$ where $1 \leq \|\mathbf{r}\|_2 \leq 2$, one can take a $\varepsilon$-net w.r.t. to Euclidean metric over all such vectors – whose size is at most $(1/\varepsilon)^{O(d \log d)}$ (see Appendix A.2) – for a small enough $\varepsilon$ so that $L_{\text{mse}}^{\text{err}}$ is essentially unaffected when $h$ is replaced by $\hat{h}$ (or vice versa) corresponding to the nearest vector in the net. For our purpose $\varepsilon$ can be $\Omega(\Delta)$.
*Net over all $D_{\text{syn}}^{\text{sup}}$*: For this we first observe that $D_{\text{syn}}^{\text{sup}}$ can always be replaced by a subset of size $s = d$ using Lemma D.2 so that $L_{\text{mse}}^{\text{err}}$ remains lower bounded by $\Delta/O(d^2)$. Thus, it suffices to consider the net $(\mathcal{N})^s$ over all $s$-sized datasets where $\mathcal{N}$ is a $\varepsilon'$-net over the set of all possible datapoints, so that $L_{\text{mse}}^{\text{err}}$ is approximately preserved by the nearest dataset in $(\mathcal{N})^s$.
The size of $(\mathcal{N})^s$ is $((B + b)/\Delta)^{O(sd)}$, and the product of the sizes of the nets constructed above is exponential in $O\left(d^2 \log(d(B+b)/\Delta)\right)$. Thus, to obtain the statement of the theorem via a union bound, the number of sampled regressors required is $k = O\left(d^2 \log(d(B+b)/\Delta) \log(1/\delta)\right)$.

**Proof Outline of Theorem 4.2.** The proof of the lower bound of $\Omega(d^2)$ on the sampled linear regressors relies on the fact that the set of symmetric $q \times q$ matrices, where $q = d+1$, is a $q(q+1)/2$-dimensional linear space. Thus, for a choice of less than $q(q+1)/2$ homogeneous linear regressors, there is a non-zero symmetric matrix $\mathbf{A}$ for which $\langle \mathbf{vv}^{\mathsf{T}}, \mathbf{A} \rangle = 0$ where $\mathbf{v}$ is any one of the chosen regression vectors. Here, $\mathbf{A}$ can be scaled so that its operator norm is exactly $1/2$. Thus, one can choose (up to appropriate scaling) $D_{\text{train}}^{\text{sup}}$ so that $\mathbb{E}_{\mathbf{z} \in D_{\text{train}}^{\text{sup}}}[\mathbf{xx}^{\mathsf{T}}] = \mathbf{I}$ which is independent of the chosen regressors. $D_{\text{syn}}^{\text{sup}}$ is then chosen so that $\mathbb{E}_{\mathbf{z} \in D_{\text{syn}}^{\text{sup}}}[\mathbf{xx}^{\mathsf{T}}] = \mathbf{I} + \mathbf{A}$ which is psd. It can be seen that each chosen regressor has the same MSE loss on $D_{\text{train}}^{\text{sup}}$ and $D_{\text{syn}}^{\text{sup}}$, while due to $\mathbf{A} \neq \mathbf{0}$, there is a regressor which has different losses on $D_{\text{train}}^{\text{sup}}$ and $D_{\text{syn}}^{\text{sup}}$.

Due to lack of space we defer the proof outlines of Theorems 4.3 and 4.4 to Appendix B.

# 6 EXPERIMENTAL EVALUATIONS

We generate synthetic datasets, $D_{\text{syn}}$ of sizes much smaller than the original training dataset using Algorithms 1 and 2 for supervised regression and offline RL respectively. We evaluate the models trained on them, over the test split of the original dataset in case of supervised regression or the derived policy in the RL environment.
**Baselines**. The following baseline datasets are included as part of our experiments. *Full Original*: model trained on the entire original training dataset $D_{\text{train}}$, and *Random*: model trained on a random sub-sample (of same size as the synthetic dataset) of the original dataset, $D_{\text{rand}}$. In addition, for supervised regression, *Leveraged*: model trained on a leverage score subsample (of same size as the synthetic dataset) of the original dataset, $D_{\text{lev}}$ (see Drineas et al. (2006)).

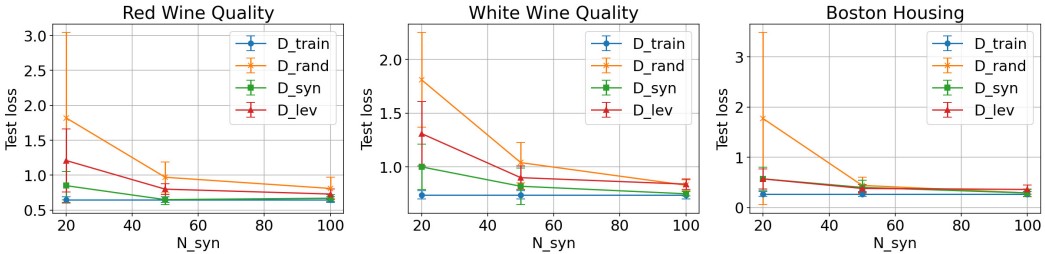

Figure 1: MSE Test losses for Homogeneous Linear Regression on the following supervised datasets: *red wine quality, white wine quality, Boston Housing.*

**Supervised Regression Datset Distillation.** We evaluate Algorithm 1 along with the above mentioned baselines on the Wine Quality (Cortez et al. (2009) ODbL 1.0 License), specifically the included red wine and white wine quality datasets, and the Boston Housing Dataset (Harrison & Rubinfeld, 1978).

The sizes of the synthetic dataset $N_{\text{syn}}$ we consider are $\{20, 50, 100\}$. The size of $D_{\text{rand}}$ and $D_{\text{lev}}$ are the same as $D_{\text{syn}}$, and is subsampled randomly from $D_{\text{train}}$. We initialise the convex optimisation of $D_{\text{syn}}$ with the initial value $D_{\text{rand}}$. We take $k = 100$ random homogeneous linear regressors in Algorithm 1 to find $D_{\text{syn}}$ and then train a homogenous linear model ($f(\mathbf{x}) = \mathbf{r}^T \mathbf{x}$) on the four datasets.

The mean test loss of models trained on respective datasets (over 10 trials) are plotted in Figure 1. Further details of the model training, hyperparameter search are included in Appendix H.1 and Appendix H.2. Experiments on larger datasets are also included in Appendix H.3.

We observe that the homogenous linear models trained on $D_{\text{syn}}$ performs almost as well as ones trained on $D_{\text{train}}$, far better than ones trained on $D_{\text{rand}}$, and better or on par with the models trained on $D_{\text{lev}}$. This demonstrates the efficacy of our synthetic data generation technique for supervised learning datasets and empirically verifies Theorem 4.1. We also observe that we perform better than the classical data-reduction technique of leverage score subsampling for homogenous linear regressors.

**Offline RL Experiments.** We test Algorithm 2 for offline RL DD by evaluating a policy trained on our synthetic dataset using Fitted-Q Iteration (Ernst et al., 2005), a classical offline RL algorithm, along with the Full Original and Random baselines on the Cartpole environment (Towers et al., 2024) (MIT License), Mountain Car MDP (Moore (1990), Towers et al. (2024) MIT License), and the Acrobot Environment (Sutton (1995), Towers et al. (2024) MIT License). Refer to Appendices H.4,H.6, H.7 for further details on the datasets. Since linear $Q$-value predictors are known to perform poorly with Fitted-Q iteration even when trained on $D_{\text{train}}$, we use 2-layer neural networks for training and generating the synthetic data which are randomly sampled by sampling the model weights independently from a standard Gaussian distribution.

We do not use $D_{\text{lev}}$ as a baseline because leverage score subsampling is only defined for linear regression and has no analog in neural network regression. To generate $D_{\text{train}}$, we sample from transitions from random(or nearly random) policies. We sample $k = 20$ random models with random normal weights for the data distillation procedure. The size of $D_{\text{rand}}$ and $D_{\text{syn}}$ is denoted by $N_{\text{syn}}$. We generate $D_{\text{syn}}$ once and we train all three datasets with the Fitted-Q iteration algorithm to get the trained policies. We evaluate these trained policies in the environment 10 times and plot our results in Figure 2. $D_{\text{rand}}$ is sampled 10 times and each sample is evaluated once. Further experiments are conducted with more values of $k$ and can be found in Appendices H.4, H.6, H.7 along with more experimental details.

We observe that a model trained on $D_{\text{syn}}$ is significantly better than one trained on $D_{\text{rand}}$. We also see that as $N_{\text{syn}}$ increases, the model trained on $D_{\text{syn}}$ performs even better or on par with one trained on $D_{\text{train}}$ for the Cartpole and Acrobot environments. This demonstrates the efficacy of our synthetic data generation technique for offline RL datasets for non-linear predictors. Additional discussions and experimental evaluations such as comparisons of our method against behavior cloning distillation baselines (Light et al., 2024) are included in Appendix H.

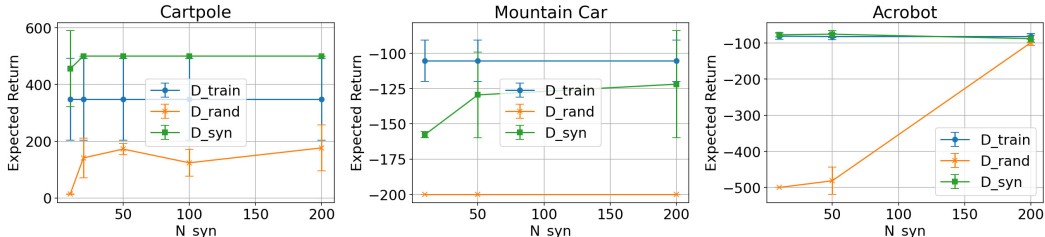

Figure 2: Max Evaluation Return with two-layer neural networks on the offline RL datasets created by the following environments: *Cartpole, Mountain Car, Acrobot.*

**Limitations.** Despite the success found in the toy sequential decision problems above, our algorithm does not yield similar performance on established offline RL benchmarks such as D4RL. The D4RL benchmark is created to distinguish between standard offline RL algorithms like *Fitted-Q iteration* (FQI) which does iterative Bellman Loss minimization, and recent advanced algorithms like *conservative Q-learning* (CQL) (see Kumar et al. (2020) and page 2 of (Fu et al., 2020)). While we propose a novel offline RL dataset distillation algorithm based on Bellman Loss matching, since D4RL is unsuited for Bellman Loss based algorithms, one would need to adapt dataset distillation to more complicated loss functions like that used in CQL. These generalizations of our techniques, their analysis and evaluations on D4RL benchmarks are beyond the scope of this paper and are deferred to future work.

**Experimental Code.** The experimental code is available at `https://github.com/google-deepmind/loss_matching_dataset_distillation`. The implementations of the algorithms in this paper are in python using the TensorFlow library. Our experiments were run on a system with standard 8-core CPU, 64GB of memory with one 16 GB RAM GPU.

## 7 CONCLUSIONS

We propose a loss matching based algorithm for supervised dataset distillation, in which given a training dataset the synthetic dataset is optimized with respect to a fixed set of randomly sampled models. For linear regression in $\mathbb{R}^d$, we prove rigorous theoretical guarantees, showing that optimizing the convex loss matching objective for only $\tilde{O}(d^2)$ sampled regressors, without any model training, suffices to obtain a high quality synthetic dataset. We prove a matching lower bound of $\Omega(d^2)$ many sampled regressors, showing the tightness of our analysis. We extend our approach to offline RL to provide an algorithm for dataset distillation matching Bellman Loss using sampled $Q$-value predictors, while showing similar performance bounds, albeit requiring $\exp(O(d \log d))$ sampled predictors in the worst case. However, under a natural decomposability assumption on $d$-dimensional state-action embeddings, we improve the upper bound to $\tilde{O}(d^2)$. Our experiments show that our algorithms yield performance gains on real datasets, both in terms of the size and quality of the synthetic dataset, even with a small number of sampled predictors.

An interesting yet challenging future direction is to extend our theoretical results in the supervised setting to neural networks, and to obtain provably efficient offline RL dataset distillation algorithms for more general classes of state-action embeddings, perhaps leveraging specific geometric properties of the associated MDPs.

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

## A  PRELIMINARIES FOR APPENDIX

### A.1  FUNCTION CLASSES AND PSEUDO-DIMENSION

We will consider a class $\mathcal{F}$ of real-valued functions (regressors) mapping $\mathbb{R}^d$ to $[-1, 1]$. The pseudo-dimension of the function class $\mathsf{Pdim}(\mathcal{F})$ is defined as the minimum cardinality subset of $\mathbb{R}^d$ which is *pseudo-shattered*. It is formally defined in Definition 11.2 of (Anthony & Bartlett, 2009).

For $\boldsymbol{\mathcal{X}} \subseteq \mathbb{R}^d$ where $|\boldsymbol{\mathcal{X}}| = N$, set $\mathcal{C}_p(\xi, \mathcal{F}, \boldsymbol{\mathcal{X}})$ to be a minimum sized $\ell_p$-metric $\xi$-cover of $\mathcal{F}$ over $\boldsymbol{\mathcal{X}}$, i.e. $\mathcal{C}_p(\xi, \mathcal{F}, \boldsymbol{\mathcal{X}})$ is a smallest subset of $\mathcal{F}$ such that for any $f^* \in \mathcal{F}$, there exists $f \in \mathcal{C}_p(\xi, \mathcal{F}, \boldsymbol{\mathcal{X}})$ s.t. $\left(\mathbb{E}_{\mathbf{x} \in \boldsymbol{\mathcal{X}}}\left[|f^*(\mathbf{x}) - f(\mathbf{x})|^p\right]\right)^{1/p} \leq \xi$ for $p \in [1, \infty)$, and $\max_{\mathbf{x} \in \boldsymbol{\mathcal{X}}} |f^*(\mathbf{x}) - f(\mathbf{x})| \leq \xi$ for $p = \infty$.

The largest size of such a cover over all choices of $\boldsymbol{\mathcal{X}} \subseteq \mathbb{R}^d$ s.t. $|\boldsymbol{\mathcal{X}}| = N$ is defined to be $N_p(\xi, \mathcal{F}, N)$.

The *pseudo-dimension* of $\mathcal{F}$, $\mathsf{Pdim}(\mathcal{F})$ (see Sec. 10.4 and 12.3 of (Anthony & Bartlett, 2009)) can be used to bound the size of covers for $\mathcal{F}$ as follows:

$$N_1(\xi, \mathcal{F}, N) \leq N_\infty(\xi, \mathcal{F}, N) \leq (eN/\xi p)^p \tag{12}$$

where $p = \mathsf{Pdim}(\mathcal{F})$ and $N \geq d$. By normalizing, the above bounds can be adapted to functions which map to $[-B, B]$ for $B > 0$.

The following theorem follows from Theorem 11.4 from (Anthony & Bartlett, 2009).

**Theorem A.1.** *The class of linear regressors over $\mathbb{R}^d$ given by $\mathbf{r}^\mathsf{T}\mathbf{x}$ for $\mathbf{r} \in \mathbb{R}$ has pseudo-dimension $d$.*

### A.2  COVER OVER $\mathcal{B}(d, r)$

Let $\mathcal{T}(d, r, \varepsilon)$ be the smallest subset of the Euclidean ball of radius $r$ in $d$-dimensions $\mathcal{B}(d, r)$ such that for all $\mathbf{r} \in \mathcal{B}(d, r)$ there exists $\hat{\mathbf{r}} \in \mathcal{T}(d, r, \varepsilon)$ s.t. $\|\mathbf{r} - \hat{\mathbf{r}}\|_2 \leq \varepsilon$. The following lemma follows from Corollary 4.2.13 of (Vershynin, 2018).

**Lemma A.2.** $\mathcal{T}(d, r, \varepsilon) \leq \left(1 + \frac{2r}{\varepsilon}\right)^d$.

### A.3  CARBERY-WRIGHT ANTI-CONCENTRATION BOUND

The following non-trivial anti-concentration of polynomials over Gaussian variables was proved by (Carbery & Wright, 2001).

**Theorem A.3** (Theorem 8 from (Carbery & Wright, 2001) )**.** *There is an absolute constant $C > 0$ such that if $f$ is any degree-$d$ polynomial over iid $N(0, 1)$ variables, then $\Pr\left[|f| \leq \varepsilon\mathbb{E}[|f|]\right] \leq Cd\varepsilon^{1/d}$ for all $\varepsilon > 0$.*

### A.4  LIPSCHITZNESS OF FEATURE-MAP $\phi$

For the our results on offline RL dataset distillation, as mentioned in Sec. 4, we assume that the feature-map $\phi$ is $L$-Lipschitz, specifically w.r.t. the $\ell_2$-metric. In other words, for any $(s_1, a_1), (s_2, a_2) \in \mathcal{B}(d_0, B_0) \times \mathcal{B}(d_0, B_0)$,

$$\|\phi(s_1, a_1) - \phi(s_2, a_2)\|_2 \leq L\|(s_1, a_1) - (s_2, a_2)\|_2 = L\sqrt{\|s_1 - s_2\|_2^2 + \|a_1 - a_2\|_2^2}$$
$$\leq L\left(\|s_1 - s_2\|_2 + \|a_1 - a_2\|_2\right)$$

### A.5  LOW RANK APPROXIMATION

A key ingredient in our lower bound is the classical Eckart–Young–Mirsky theorem (Eckart & Young, 1936; Stewart, 1993), which states that for any symmetric positive semidefinite matrix $\mathbf{A} \in \mathbb{R}^{d \times d}$ with eigenvalues $\lambda_1 \geq \lambda_2 \geq \cdots \geq \lambda_d$, the best rank-$m$ approximation $\mathbf{B}$ in spectral norm is obtained by truncating the eigendecomposition of $\mathbf{A}$, and the approximation error is exactly

$$\min_{\mathrm{rank}(\mathbf{B}) \leq m} \|\mathbf{A} - \mathbf{B}\|_2 = \lambda_{m+1}. \tag{13}$$

This immediately implies that if $\lambda_{m+1} \geq \varepsilon$, then any rank-$m$ factorization $\mathbf{B} = \mathbf{Z}\mathbf{Z}^\top$ with $\mathbf{Z} \in \mathbb{R}^{d \times m}$ must incur spectral error at least $\varepsilon$.

### A.6 Chernoff Bound

We also use the following well known concentration bound.

**Theorem A.4** (Chernoff Bound). *Let $X_1, \ldots, X_n$ be iid $\{0, 1\}$-valued random variables and let $\mu = \mathbb{E}\left[\sum_{i=1}^n X_i\right]$. Then for any $\delta > 0$,*

$$\Pr\left[\sum_i^n X_i \leq (1 - \delta)\mu\right] \leq \exp\left(-\delta^2 \mu / 2\right).$$

## B Proof Outlines Contd.

**Proof Outline of Theorem 4.3.** The main complication as compared to the supervised regression case is the presence of the $\max$ term in the Bellman loss, which we circumvent using a conditioning argument. Specifically, consider some action-value predictor $f \in \mathcal{Q}_0$ given by $f(s, a) := \mathbf{r}^\top \phi(s, a)$ such that $\hat{L}_{\text{Bell}}^{\text{err}}(D_{\text{train}}^{\text{orl}}, D_{\text{syn}}^{\text{orl}}, f, 1) > \varepsilon$, for some $D_{\text{syn}}^{\text{orl}}$. Assume for ease of exposition that $\|\mathbf{r}\|_2 = 1$. Now, a $\hat{f} \sim H$ corresponds to a vector $\hat{\mathbf{r}} = \alpha \mathbf{r} + \mathbf{J}\mathbf{u}$ where $\alpha \sim N(0, 1)$, $\mathbf{u} \sim N(0, 1)^{d-1}$ and $\mathbf{J}$ is a $d \times (d - 1)$ matrix with columns being a completion of $\mathbf{r}$ to an orthonormal basis. It is easy to see that with probability at least $\exp(-O(d \log d))$, $\|\mathbf{u}\|_2 \ll 1$ and therefore $\|\mathbf{J}\mathbf{u}\|_2 \ll 1$. Conditioning on this event and the positivity of $\alpha$ allows us to use $\max_{a'} \hat{f}(\phi(s', a')) \approx \alpha \max_{a'} f(\phi(s', a'))$. This, along with a conditioning on $\lambda \approx 1$ and some algebraic manipulations yields that $\hat{L}_{\text{Bell}}^{\text{err}}(D_{\text{train}}^{\text{orl}}, D_{\text{syn}}^{\text{orl}}, \hat{f}) \approx \alpha^4 \Delta$, which directly yields a probabilistic lower bound of $\Omega(\Delta)$ on $\hat{L}_{\text{Bell}}^{\text{err}}(D_{\text{train}}^{\text{orl}}, D_{\text{syn}}^{\text{orl}}, \hat{f})$. The rest of the net based arguments are analogous to the supervised case but with notable differences. In particular, we observe that $D_{\text{syn}}^{\text{orl}}$ can always be replaced by a subset of size $s = O(d/\Delta^2 \log(1/\Delta))$ so that $L_{\text{Bell}}^{\text{err}}$ remains lower bounded by $O(\Delta)$. In addition, we use a more broadly applicable generalization error bound for the Bellman loss using its pseudo-dimension and the Lipschitzness of $\phi$. However, due to the $\exp(-O(d \log d))$ of the conditioning, success probability for a single sample $(\hat{f}, \lambda)$ is also $\exp(-O(d \log d))$ and therefore the number of predictors to be sampled is $\exp(O(d \log d))$.

**Proof outline of Theorem 4.4.** We observe that when $\phi$ is decomposable, for a $Q$-value predictor $f(s, a) := \mathbf{v}^\top \phi(s, a)$, we have that $\max_{a' \in \mathcal{A}} f(s', a') = \mathbf{v}^\top \phi_1(s') + \max_{a' \in \mathcal{A}} \mathbf{v}^\top \phi_2(a')$. Here $\alpha := \max_{a' \in \mathcal{A}} \mathbf{v}^\top \phi_2(a')$ depends only on $\mathbf{v}$ and is independent of the dataset. This allows for cancellations of terms between $D_{\text{train}}^{\text{orl}}$ and $D_{\text{syn}}^{\text{orl}}$ in $\hat{L}_{\text{Bell}}^{\text{err}}$, and using the constraint in equation 11 we obtain a linear regression formulation in the embedding space $\mathbb{R}^d$. Thus, one can essentially follow the proof of Theorem 4.1. Further, it is easy to see that if $\phi_1$ and $\phi_2$ are linear then the constraint in equation 11 is linear in the points of $D_{\text{syn}}^{\text{orl}}$ resulting in a convex optimization.

## C Proof of Theorem 4.1

$D_{\text{train}}^{\text{sup}} = \{(\mathbf{x}_i, y_i) \in \mathcal{B}(d, B) \times [-b, b]\}_{i=1}^n$ is the given training dataset consisting of feature-vector and real-valued label pairs for a regression task. For ease of notation in the proof of this theorem we shall drop the concatenation operator $\zeta$ and instead use vectors to denote the data point with the feature-vector and label concatenated. Further, we let $q := d + 1$ represent the dimensionality so that the domain of $\mathcal{F}$ is $\mathbb{R}^q$. With this notation, $D_{\text{train}}^{\text{sup}} = \{\mathbf{x}_i \in \mathcal{B}(q - 1, B) \times [-b, b]\}_{i=1}^n$ is the given training dataset. We use an analogous notation for the datapoints of the synthetic datasets in the analysis below.

### C.1 Bounds for fixed $\hat{h}$ and fixed synthetic dataset $\hat{D}$

We begin by fixing (i) $\hat{h} \in \mathcal{F}$ s.t.

$$\hat{h}(\mathbf{x}) := \hat{c}\mathbf{u}^\top \mathbf{x} \text{ for some } \mathbf{u} \in \mathbb{R}^q \text{ s.t. } \|\mathbf{u}\|_2 = 1 \text{ and } \hat{c} \in [1, 2] \tag{14}$$

and, $\hat{D} = \{\hat{\mathbf{z}}_i \in \mathcal{B}(q-1, B) \times [-b, b]\}_{i=1}^s$ for some $s \in \mathbb{Z}^+$. Using this, we have

$$L_{\text{mse}}^{\text{err}}(D_{\text{train}}^{\text{sup}}, \hat{D}, \hat{h}) = \left( \frac{1}{n} \sum_{i=1}^n (\hat{c}\mathbf{u}^\mathsf{T}\mathbf{x}_i)^2 - \frac{1}{s} \sum_{i=1}^s (\hat{c}\mathbf{u}^\mathsf{T}\hat{\mathbf{z}}_i)^2 \right)^2 \tag{15}$$

The following lemma provides a key probabilistic lower bound.

**Lemma C.1.** *Let* $L_{\text{mse}}^{\text{err}}(D_{\text{train}}^{\text{sup}}, \hat{D}, \hat{h}) \geq \Delta$. *Then, for a randomly chosen* $g \sim G(\mathcal{F})$ *s.t.* $g(\mathbf{x}) := \mathbf{r}^\mathsf{T}\mathbf{x}$, *we have*

$$\Pr\left[ \left( L_{mse}^{\text{err}}(D_{\text{train}}^{\text{sup}}, \hat{D}, g) \geq \frac{\Delta}{C_0 q^2} \right) \bigwedge \left( \|\mathbf{r}\|_2^2 \leq 10 \right) \right] \geq \frac{7}{30}$$

*for some absolute constant* $C_0 > 0$.

*Proof.* We first lower bound the expectation $\mathbb{E}\left[ L_{\text{mse}}^{\text{err}}(D_{\text{train}}^{\text{sup}}, \hat{D}, g) \right]$. Sampling a random $g \in G$ corresponds to sampling $\mathbf{r} \sim N(0, 1/q)^q$ u.a.r. and letting $g(\mathbf{x}) := \mathbf{r}^\mathsf{T}\mathbf{x}$ for $\mathbf{x} \in \mathbb{R}^q$. Letting $\mathbf{r}_1 := \mathbf{u}$, we can find unit vectors $\mathbf{r}_2, \ldots, \mathbf{r}_q$ such that $\mathbf{r}_1, \ldots, \mathbf{r}_q$ is an orthonormal basis. Writing $\mathbf{r} := \alpha_1 \mathbf{r}_1 + \cdots + \alpha_q \mathbf{r}_q$, we obtain that $\alpha_j$ is iid $N(0, 1/q)$ for each $j \in q$.

Therefore,

$$L_{\text{mse}}^{\text{err}}(D_{\text{train}}^{\text{sup}}, \hat{D}, g) = \left( \frac{1}{n} \sum_{i=1}^n \left( \sum_{j=1}^q \alpha_j \mathbf{r}_1^\mathsf{T}\mathbf{x}_i \right)^2 - \frac{1}{s} \sum_{i=1}^s \left( \sum_{j=1}^q \alpha_j \mathbf{r}_j^\mathsf{T}\hat{\mathbf{z}}_i \right)^2 \right)^2 \tag{16}$$

Define $\beta_i = \sum_{j=2}^q \alpha_j \mathbf{r}_j^\mathsf{T}\mathbf{x}_i$ for $i \in [n]$ and similarly $\gamma_j = \sum_{j=2}^q \alpha_j \mathbf{r}_j^\mathsf{T}\hat{\mathbf{z}}_i$ for $i \in [s]$. Note that both $\{\beta_i\}_{i=1}^n, \{\gamma_i\}_{i=1}^s$ are independent of $\alpha_1$. We now rewrite the squared loss in terms of $\{\beta_i\}_{i=1}^n, \{\gamma_i\}_{i=1}^s$ to obtain:

$$L_{\text{mse}}^{\text{err}}(D_{\text{train}}^{\text{sup}}, \hat{D}, g) = \left( \frac{1}{n} \sum_{i=1}^n (\alpha_1 \mathbf{r}_1^\mathsf{T}\vec{\mathbf{x}}_i + \beta_i)^2 - \frac{1}{s} \sum_{i=1}^s (\alpha_1 \mathbf{r}_1^\mathsf{T}\hat{\mathbf{z}}_i + \gamma_i)^2 \right)^2$$

$$= \left[ \alpha_1^2 \left( \frac{1}{n} \sum_{i=1}^n (\mathbf{r}_1^\mathsf{T}\mathbf{x}_i)^2 - \frac{1}{s} \sum_{i=1}^s (\mathbf{r}_1^\mathsf{T}\hat{\mathbf{z}}_i)^2 \right) \right.$$

$$\left. + 2\alpha_1 \left( \frac{1}{n} \sum_{i=1}^n \beta_i \mathbf{r}_1^\mathsf{T}\mathbf{x}_i - \frac{1}{s} \sum_{i=1}^s \gamma_i \mathbf{r}_1^\mathsf{T}\hat{\mathbf{z}}_i \right) + \left( \frac{1}{n} \sum_{i=1}^n \beta_i^2 - \frac{1}{s} \sum_{i=1}^s \gamma_i^2 \right) \right]^2 \tag{17}$$

At this point, we let $\kappa := \left( \frac{1}{n} \sum_{i=1}^n (\mathbf{r}_1^\mathsf{T}\mathbf{x}_i)^2 - \frac{1}{s} \sum_{i=1}^s (\mathbf{r}_1^\mathsf{T}\hat{\mathbf{z}}_i)^2 \right)$, $\omega = \left( \frac{1}{n} \sum_{i=1}^n \beta_i^2 - \frac{1}{s} \sum_{i=1}^s \gamma_i^2 \right)$ and $\lambda = 2 \left( \frac{1}{n} \sum_{i=1}^n \beta_i \mathbf{r}_1^\mathsf{T}\mathbf{x}_i - \frac{1}{s} \sum_{i=1}^s \gamma_i \mathbf{r}_1^\mathsf{T}\hat{\mathbf{z}}_i \right)$. Note that $\kappa, \omega, \lambda$ are independent of $\alpha_1$. Substituting these into equation 17 we obtain

$$L_{\text{mse}}^{\text{err}}(D_{\text{train}}^{\text{sup}}, \hat{D}, g) = \left( \alpha_1^2 \kappa + \alpha_1 \lambda + \omega \right)^2$$
$$= \alpha_1^4 \kappa^2 + \alpha_1^2 \lambda^2 + \omega^2 + 2\alpha_1^3 \lambda\kappa + 2\alpha_1 \lambda\omega + 2\alpha_1^2 \omega\kappa$$

Taking the expectation over $\alpha_1$ yields,

$$\mathbb{E}_{\alpha_1}[L_{\text{mse}}^{\text{err}}(D_{\text{train}}^{\text{sup}}, \hat{D}, g)] = \mathbb{E}\left[\alpha_1^4\right]\kappa^2 + \mathbb{E}\left[\alpha_1^2\right]\lambda^2 + \omega^2 + 2\mathbb{E}\left[\alpha_1^3\right]\lambda\kappa + 2\mathbb{E}\left[\alpha_1\right]\lambda\omega + 2\mathbb{E}[\alpha_1^2]\omega\kappa$$
$$= (3/q^2)\kappa^2 + \lambda^2/q + \omega^2 + (2/q)\omega\kappa$$
$$= (2/q^2)\kappa^2 + (\kappa/q + \omega)^2 + \lambda^2/q$$
$$\geq (2/q^2)\kappa^2$$
$$= \frac{2}{q^2} \left( \frac{1}{n} \sum_{i=1}^n (\mathbf{r}_1^\mathsf{T}\mathbf{x}_i)^2 - \frac{1}{s} \sum_{i=1}^s (\mathbf{r}_1^\mathsf{T}\hat{\mathbf{z}}_i)^2 \right)^2$$
$$= 2\left( \frac{1}{q\hat{c}} \right)^2 \left[ \frac{1}{n} \sum_{i=1}^n (\hat{c}\mathbf{u}^\mathsf{T}\mathbf{x}_i)^2 - \frac{1}{s} \sum_{i=1}^s (\hat{c}\mathbf{u}^\mathsf{T}\hat{\mathbf{z}}_i)^2 \right]^2$$
$$= 2\left( \frac{1}{q\hat{c}} \right)^2 L_{\text{mse}}^{\text{err}}(D_{\text{train}}^{\text{sup}}, \hat{D}, \hat{h}) \geq \frac{\Delta}{2q^2} \tag{18}$$

where we used equation 14 and equation 15 along with the fact that $\alpha_1 \sim N(0, 1/q)$. Observe that $L_{\text{mse}}^{\text{err}}(D_{\text{train}}^{\text{sup}}, \hat{D}, g)$ is a degree-4 square polynomial in $\{\alpha_j\}_{j=1}^q$ and hence is always positive. Observe that the lower bound in equation 18 is independent of $\{\alpha_j\}_{j=2}^q$ and is thus an lower bound for the expectation over $\{\alpha_j\}_{j=1}^q$. Applying Theorem A.3 (see Appendix A.3) to equation 18 we obtain

$$\Pr\left[L_{\text{mse}}^{\text{err}}(D_{\text{train}}^{\text{sup}}, \hat{D}, g) \geq \left(\frac{1}{72}\right)^4 \frac{\Delta}{2q^2}\right] \geq \frac{1}{3}. \tag{19}$$

Further, since $\mathbb{E}[\|\mathbf{r}\|_2^2] = 1$, by Markov's inequality, $\Pr[\|\mathbf{r}\|_2^2 > 10] < 1/10$. This along with the equation 19 and taking $C_0 = (1/2)(1/72)^4$ completes the proof. $\qquad\square$

The following is a straightforward implication of Lemma C.1 along with the Chernoff Bound (Theorem A.4).

**Lemma C.2.** *Let* $L_{\text{mse}}^{\text{err}}(D_{\text{train}}^{\text{sup}}, \hat{D}, \hat{h}) > \Delta$. *Then, for iid random* $g_1, \ldots, g_k \sim G(\mathcal{F})$, *s.t.* $g_j(\mathbf{x}) := \mathbf{v}_j^\mathsf{T} \mathbf{x}$ *we have*

$$\Pr\left[\left|\left\{j \in [k] : \left(L_{\text{mse}}^{\text{err}}(D_{\text{train}}^{\text{sup}}, \hat{D}, g_j) \geq \frac{\Delta}{C_0 q^2}\right) \wedge \left(\|\mathbf{v}_j\|_2^2 \leq 10\right)\right\}\right| \geq \frac{7k}{60}\right] \geq 1 - \exp\left(\frac{-7k}{240}\right)$$

## C.2 NET OVER REGRESSORS $h$ AND SYNTHETIC DATASETS $D_{\text{syn}}^{\text{sup}}$

First we shall construct a net over regressors $h \in \mathcal{F}$, where $h(\mathbf{x}) := \mathbf{r}^\mathsf{T} \mathbf{x}$ for some $\mathbf{r} \in \mathcal{B}(q, 2)$. Thus, we can consider the cover $\mathcal{T}(q, 2, \xi))$ (see Appendix A.2) and let $\hat{\mathcal{F}} = \{\hat{h} : \exists \hat{\mathbf{r}} \in \mathcal{T}(q, 2, \xi) \text{ s.t. } \hat{h}(\mathbf{x}) := \hat{\mathbf{r}}^\mathsf{T} \mathbf{x}\}$, for some parameter $\xi \in (0, 1)$ to be chosen later. The following is a simple approximation lemma.

**Lemma C.3.** *For any* $D_{\text{train}}^{\text{sup}}$ *and* $\hat{D}$ *as defined in the previous subsection, for any* $h \in \mathcal{F}$, $\exists \hat{h} \in \hat{\mathcal{F}}$ *s.t.*

$$\left|L_{\text{mse}}^{\text{err}}(D_{\text{train}}^{\text{sup}}, \hat{D}, h) - L_{\text{mse}}^{\text{err}}(D_{\text{train}}^{\text{sup}}, \hat{D}, \hat{h})\right| \leq 65\xi(B + b)^4. \tag{20}$$

*Proof.* Let $h(\mathbf{x}) := \mathbf{r}^\mathsf{T} \mathbf{x}$, where $\mathbf{r} \in \mathcal{B}(q, 2)$. Choose $\hat{\mathbf{r}} \in \mathcal{T}(q, 2, \xi)$ s.t. $\|\mathbf{r} - \hat{\mathbf{r}}\|_2 \leq \xi$ and define $\hat{h}$ to be $\hat{h}(\mathbf{x}) := \hat{\mathbf{r}}^\mathsf{T} \mathbf{x}$. Thus, for every $\mathbf{x}_i \in D_{\text{train}}^{\text{sup}}$, $|(\mathbf{r}^\mathsf{T} \mathbf{x}_i)^2 - (\hat{\mathbf{r}}^\mathsf{T} \mathbf{x}_i)^2| \leq \left((\mathbf{r} - \hat{\mathbf{r}})^\mathsf{T} \mathbf{x}_i\right)^2 + 2\left|\left((\mathbf{r} - \hat{\mathbf{r}})^\mathsf{T} \mathbf{x}_i\right) \mathbf{r}^\mathsf{T} \mathbf{x}_i\right| \leq (\xi(B + b))^2 + 4\xi(B + b)^2 \leq 5\xi(B + b)^2$ since $\xi \leq 1$. Using this, along with equation 15 we have that $L_{\text{mse}}^{\text{err}}(D_{\text{train}}^{\text{sup}}, \hat{D}, \hat{h}) = \left(\sqrt{L_{\text{mse}}^{\text{err}}(D_{\text{train}}^{\text{sup}}, \hat{D}, h)} + \upsilon\right)^2$ where $\upsilon \in \mathbb{R}$ s.t. $|\upsilon| \leq 5\xi(B + b)^2$.

Therefore, $\left|L_{\text{mse}}^{\text{err}}(D_{\text{train}}^{\text{sup}}, \hat{D}, \hat{h}) - L_{\text{mse}}^{\text{err}}(D_{\text{train}}^{\text{sup}}, \hat{D}, h)\right| \leq 2|\upsilon|\sqrt{L_{\text{mse}}^{\text{err}}(D_{\text{train}}^{\text{sup}}, \hat{D}, h)} + |\upsilon|^2$. It is easy to see using the norm bounds on the regressor vectors and the data points along with equation 15 that $\sqrt{L_{\text{mse}}^{\text{err}}(D_{\text{train}}^{\text{sup}}, \hat{D}, h)} \leq 4(B + b)^2$. Thus, $2|\upsilon|\sqrt{L_{\text{mse}}^{\text{err}}(D_{\text{train}}^{\text{sup}}, \hat{D}, h)} + |\upsilon|^2 \leq 40\xi(B + b)^4 + 25\xi^2(B + b)^4 \leq 65\xi(B + b)^4$, which completes the proof. $\qquad\square$

We now show that the synthetic data $D_{\text{syn}}^{\text{sup}}$ can be approximated with a much smaller dataset $\hat{D}$ whose points are from an appropriate Euclidean net.

**Lemma C.4.** *Fix a* $\nu \in (0, 1)$. *For any* $D_{\text{syn}}^{\text{sup}}$, *there exists a dataset* $\hat{D}$ *of size* $s := q$ *whose points are from Euclidean net* $\mathcal{T}(q, \sqrt{q}(B + b), \nu\sqrt{q}(B + b)/2)$ *such that for any* $h \in \mathcal{F}$,

$$\left|L_{\text{mse}}^{\text{err}}(D_{\text{train}}^{\text{sup}}, \hat{D}, h) - L_{\text{mse}}^{\text{err}}(D_{\text{train}}^{\text{sup}}, D_{\text{syn}}^{\text{sup}}, h)\right| \leq 65\nu q^2(B + b)^4 \tag{21}$$

*Proof.* Let us first take $\overline{D}$ (bounded in a ball of radius $\sqrt{q}(B + b)$) to be the dataset of size $q$ from Lemma D.2 such that

$$L_{\text{mse}}^{\text{err}}(D_{\text{train}}^{\text{sup}}, \overline{D}, h) = L_{\text{mse}}^{\text{err}}(D_{\text{train}}^{\text{sup}}, D_{\text{syn}}^{\text{sup}}, h) \tag{22}$$

Then we create $\hat{D}$ by replacing each point in $\overline{D}$ with the closest point in $\mathcal{T}(q, \sqrt{q}(B+b), \sqrt{q}\nu(B+b)/2)$. Using arguments analogous to those in the proof of Lemma C.3 we obtain,

$$\left| L_{\text{mse}}^{\text{err}}(D_{\text{train}}^{\text{sup}}, \hat{D}, h) - L_{\text{mse}}^{\text{err}}(D_{\text{train}}^{\text{sup}}, \overline{D}, h) \right| \leq 65\nu q^2 (B+b)^4 \tag{23}$$

Combining the above two inequality and equation completes the proof. $\square$

Note that in In Lemma C.4 we replace the $m$-sized synthetic dataset $D_{\text{syn}}^{\text{sup}}$ with a dataset $\hat{D}$ of size $q = d + 1$ with points from an appropriate net. This allows us to take a union bound over all possible $q$-sized $\hat{D}$. The value of $m$ is unrestricted, and need not depend on $d$. There is no restriction on $n = |D_{\text{train}}^{\text{sup}}|$ either, and we in any case do not require a net argument for $D_{\text{train}}^{\text{sup}}$. Due to this net argument which replaces $D_{\text{syn}}^{\text{sup}}$ with $q$-sized $\hat{D}$ (albeit with possibly larger norms), the analysis is made independent of $m$ and thus $\varepsilon$ and $\delta$ are also independent of $m$.

### C.3 Union bound over net and completing the proof

To use the analysis in Sec. C.2, let us set $\xi = \Delta/(10^8 C_0 q^2 (B+b)^4)$ and $\nu = \Delta/(10^8 C_0 q^4 (B+b)^4)$. Applying Lemma C.4 and followed by Lemma C.3 directly yields the following combined net for the regressors $h$ and the synthetic dataset.

**Lemma C.5.** *For any $h \in \mathcal{F}$ and $D_{\text{syn}}^{\text{sup}}$, there exist $\hat{h} \in \hat{\mathcal{F}}$ and $\hat{D} \in \mathcal{T}(q, \sqrt{q}(B+b), \nu\sqrt{q}(B+b)/2)^s$ such that*

$$\left| L_{\text{mse}}^{\text{err}}(D_{\text{train}}^{\text{sup}}, \hat{D}, \hat{h}) - L_{\text{mse}}^{\text{err}}(D_{\text{train}}^{\text{sup}}, D_{\text{syn}}^{\text{sup}}, h) \right| \leq \Delta/(10^6 C_0 q^2 (B+b)^4) \tag{24}$$

*where $s := q$ for some constant $C_1$ and $\hat{\mathcal{F}} = \{\hat{h} : \exists \hat{\mathbf{r}} \in \mathcal{T}(q, 2, \xi) \text{ s.t. } h(\mathbf{x}) := \hat{\mathbf{r}}^{\mathsf{T}}\mathbf{x}\}$.*

Observe that

$$\left| \hat{\mathcal{F}} \times \mathcal{T}(q, \sqrt{q}(B+b), \nu\sqrt{q}(B+b)/2)^s \right| \leq \left( \frac{6}{\xi} \right)^q \cdot \left( \frac{6}{\nu} \right)^{qs} = \exp\left( O\left( q^2 \log\left( \frac{q(B+b)}{\Delta} \right) \right) \right)$$

Thus, one can apply Lemma C.2 with $k = O\left( q^2 \log\left( \frac{q(B+b)}{\Delta} \right) \log\left( \frac{1}{\delta} \right) \right)$ and take a union bound over $\hat{\mathcal{F}} \times \mathcal{T}(q, \sqrt{q}(B+d), \nu\sqrt{q}(B+b)/2)^s$ to obtain the following: with probability $(1 - \delta)$ over iid random $g_1, \ldots, g_k \sim G$, if $L_{\text{mse}}^{\text{err}}(D_{\text{train}}^{\text{sup}}, \hat{D}, \hat{h}) > \Delta$ then there exist $7k/60$ distinct $j^* \in [k]$ s.t.,

$$\left( L_{\text{mse}}^{\text{err}}(D_{\text{train}}^{\text{sup}}, \hat{D}, g_{j^*}) \geq \frac{\Delta}{C_0 q^2} \right) \wedge \left( \|\mathbf{v}_{j^*}\|_2^2 \leq 10 \right) \tag{25}$$

Now, since $\|\mathbf{v}_{j^*}\|_2^2 \leq 10$, $\tilde{c}g_{j^*} \in \mathcal{F}$ for some $\tilde{c} \geq 2/\sqrt{10}$. And since $L_{\text{mse}}^{\text{err}}(D_{\text{train}}^{\text{sup}}, \hat{D}, g_{j^*})$ is proportional to $\|\mathbf{v}_{j^*}\|_2^4$, applying Lemma C.4 to $L_{\text{mse}}^{\text{err}}(D_{\text{train}}^{\text{sup}}, \hat{D}, \tilde{c}g_{j^*})$ yields that

$$L_{\text{mse}}^{\text{err}}(D_{\text{train}}^{\text{sup}}, D_{\text{syn}}^{\text{sup}}, g_{j^*}) \geq \frac{\Delta}{2C_0 q^2}$$

which implies,

$$\sum_{j=1}^{k} L_{\text{mse}}^{\text{err}}(D_{\text{train}}^{\text{sup}}, D_{\text{syn}}^{\text{sup}}, g_j) \geq \frac{7k\Delta}{120 C_0 q^2}$$

completing the proof of Theorem 4.1.

## D Lower Bounds for Supervised Learning Dataset Distillation

$D_{\text{train}}^{\text{sup}} = \{(\mathbf{x}_i, y_i) \in \mathcal{B}(d, B) \times [-b, b]\}_{i=1}^n$ is the given training dataset consisting of feature-vector and real-valued label pairs for a regression task and our goal is to find the synthetic dataset $D_{\text{syn}}^{\text{sup}} = \{(\mathbf{z}_i, \hat{y}_i) \in \mathcal{B}(d, B) \times [-b, b]\}_{i=1}^m$. For ease of notation in the proof of this theorem we shall drop the concatenation operator $\zeta$ and instead use vectors to denote the data point with the feature-vector and label concatenated. Further, we let $q := d + 1$ represent the dimensionality so that the domain of $\mathcal{F}$ is $\mathbb{R}^q$. With this notation, $D_{\text{train}}^{\text{sup}} = \{\mathbf{x}_i \in \mathcal{B}(q - 1, B) \times [-b, b]\}_{i=1}^n$ is the given

training dataset. We use an analogous notation for the datapoints of the synthetic datasets in the analysis below.

We begin by fixing $f \in \mathcal{F}$ s.t.

$$f(\mathbf{x}) := \hat{c}\mathbf{u}^\mathsf{T}\mathbf{x} \text{ for some } \mathbf{u} \in \mathbb{R}^q \text{ s.t. } \|\mathbf{u}\|_2 = 1 \text{ and } \hat{c} \in [1, 2]$$

and, $D_{\text{syn}}^{\text{sup}} = \{\mathbf{z}_i \in \mathcal{B}(q-1, B) \times [-b, b]\}_{i=1}^s$ for some $s \in \mathbb{Z}^+$. Further, let us also define $\mathbf{X} = [\mathbf{x}_1, \mathbf{x}_2, \ldots, \mathbf{x}_n] \in \mathbb{R}^{q \times n}$ and $\mathbf{Z} = [\mathbf{z}_1, \mathbf{z}_2, \ldots, \mathbf{z}_m] \in \mathbb{R}^{q \times m}$.

Using this, we have

$$
\begin{aligned}
L_{\text{mse}}^{\text{err}}(D_{\text{train}}^{\text{sup}}, D_{\text{syn}}^{\text{sup}}, f) &= \left( \frac{1}{n}\sum_{i=1}^n (\hat{c}\mathbf{u}^\mathsf{T}\mathbf{x}_i)^2 - \frac{1}{m}\sum_{i=1}^m (\hat{c}\mathbf{u}^\mathsf{T}\mathbf{z}_i)^2 \right)^2 \\
&= \hat{c}^4 \left( \frac{1}{n}\sum_{i=1}^n \mathbf{u}^\mathsf{T}\mathbf{x}_i\mathbf{x}_i^\mathsf{T}\mathbf{u} - \frac{1}{m}\sum_{i=1}^m \mathbf{u}^\mathsf{T}\mathbf{z}_i\mathbf{z}_i^\mathsf{T}\mathbf{u} \right)^2 \\
&= \hat{c}^4 \left( \mathbf{u}^\mathsf{T}\left( \frac{1}{n}\sum_{i=1}^n \mathbf{x}_i\mathbf{x}_i^\mathsf{T} \right)\mathbf{u} - \mathbf{u}^\mathsf{T}\left( \frac{1}{m}\sum_{i=1}^m \mathbf{z}_i\mathbf{z}_i^\mathsf{T} \right)\mathbf{u} \right)^2 \\
&= \hat{c}^4 \left( \mathbf{u}^\mathsf{T}\left( \frac{1}{n}\sum_{i=1}^n \mathbf{x}_i\mathbf{x}_i^\mathsf{T} - \frac{1}{m}\sum_{i=1}^m \mathbf{z}_i\mathbf{z}_i^\mathsf{T} \right)\mathbf{u} \right)^2 \\
&= \hat{c}^4 \left( \mathbf{u}^\mathsf{T}\left( \frac{\mathbf{X}\mathbf{X}^\mathsf{T}}{n} - \frac{\mathbf{Z}\mathbf{Z}^\mathsf{T}}{m} \right)\mathbf{u} \right)^2
\end{aligned}
\tag{26}
$$

**Lemma D.1.** *Let $\mathbf{Z} \in \mathbb{R}^{q \times m}$ have columns $\mathbf{z}_j = (\mathbf{u}_j, s_j)$ with $\mathbf{u}_j \in \mathbb{R}^{q-1}$, $s_j \in \mathbb{R}$, satisfying $\|\mathbf{u}_j\|_2 \leq B$ and $|s_j| \leq b$ for all $j$. Let $\overline{\mathbf{Z}} \in \mathbb{R}^{q \times q}$ have columns $\overline{\mathbf{z}}_i = (\mathbf{v}_i, t_i)$ and satisfy $\frac{1}{m}\mathbf{Z}\mathbf{Z}^\top = \frac{1}{q}\overline{\mathbf{Z}}\,\overline{\mathbf{Z}}^\top$. Then for all $i$, $\|\mathbf{v}_i\|_2 \leq \sqrt{q}\,B$ and $|t_i| \leq \sqrt{q}\,b$.*

*Proof.* Write the block decomposition $\frac{1}{m}\mathbf{Z}\mathbf{Z}^\top = \frac{1}{q}\overline{\mathbf{Z}}\,\overline{\mathbf{Z}}^\top = \begin{pmatrix} \mathbf{A} & \mathbf{c} \\ \mathbf{c}^\top & \alpha \end{pmatrix}$ with $\mathbf{A} = \frac{1}{m}\sum_{j=1}^m \mathbf{u}_j\mathbf{u}_j^\top$ and $\alpha = \frac{1}{m}\sum_{j=1}^m s_j^2$. Let $\mathbf{V} \in \mathbb{R}^{(q-1) \times q}$ have columns $\mathbf{v}_i$. Then $\frac{1}{q}\mathbf{V}\mathbf{V}^\top = \mathbf{A}$, so $\|\mathbf{V}\|_2^2 \leq q\,\text{tr}(\mathbf{A}) \leq qB^2$, giving $\|\mathbf{v}_i\|_2 \leq \sqrt{q}\,B$, for all $i = 1, \ldots, q$. Similarly, for $t = (t_1, \ldots, t_q)$ we have $\frac{1}{q}\sum_i t_i^2 = \alpha \leq b^2$, giving $|t_i| \leq \sqrt{q}\,b$, for all $i = 1, \ldots, q$. $\square$

As $\mathbf{Z}\mathbf{Z}^\mathsf{T} \in \mathbb{R}^{q \times q}$ is a positive semi-definite matrix, we observe that by using the Cholesky factorization the synthetic data $D_{\text{syn}}^{\text{sup}}$ (corresponding to $\mathbf{Z} \in \mathbb{R}^{q \times m}$) can always be replaced by another dataset $\overline{D}$ (corresponding to $\overline{\mathbf{Z}} \in \mathbb{R}^{q \times q}$) of size $q$ such that $\mathbf{Z}\mathbf{Z}^\mathsf{T}/m = \overline{\mathbf{Z}}\overline{\mathbf{Z}}^\mathsf{T}/q$ without any change in the loss. Together with the norm bounds on the data points of $\overline{D}$ from the lemma above, this leads to the following lemma which we use in Appendix C:

**Lemma D.2.** *For any $D_{\text{syn}}^{\text{sup}} \in (\mathcal{B}(d, B) \times [-b, b])^m$ and $D_{\text{train}}^{\text{sup}} \in (\mathcal{B}(d, B) \times [-b, b])^n$, there exists a dataset $\overline{D} \in (\mathcal{B}(d, B\sqrt{d+1}) \times [-b\sqrt{d+1}, b\sqrt{d+1}])^q$ of size $d + 1$ such that $L_{\text{mse}}^{\text{err}}(D_{\text{train}}^{\text{sup}}, D_{\text{syn}}^{\text{sup}}, f) = L_{\text{mse}}^{\text{err}}(D_{\text{train}}^{\text{sup}}, \overline{D}, f)$ for any $f \in \mathcal{F}$.*

## D.1 LOWER BOUND ON THE SIZE OF SYNTHETIC DATA

Given a parameter $\varepsilon$ and a dataset $D_{\text{train}}^{\text{sup}}$ (corresponding to $\mathbf{X} \in \mathbb{R}^{q \times n}$), the objective of the supervised learning dataset distillation problem to find a synthetic dataset $D_{\text{syn}}^{\text{sup}}$ (corresponding to $\mathbf{Z} \in \mathbb{R}^{q \times m}$) such that $L_{\text{mse}}^{\text{err}}(D_{\text{train}}^{\text{sup}}, D_{\text{syn}}^{\text{sup}}, f) \leq \varepsilon$ for all $f \in \mathcal{F}$, as stated in equation 1.

This means that in equation 26, we need to have $L_{\text{mse}}^{\text{err}}(D_{\text{train}}^{\text{sup}}, D_{\text{syn}}^{\text{sup}}, f) = \hat{c}^4 \left( \mathbf{u}^\mathsf{T}\left( \frac{\mathbf{X}\mathbf{X}^\mathsf{T}}{n} - \frac{\mathbf{Z}\mathbf{Z}^\mathsf{T}}{s} \right)\mathbf{u} \right)^2 \leq \varepsilon$ for all unit vectors $\mathbf{u} \in \mathbb{R}^q$. This implies that the matrix corresponding to synthetic dataset $Z$ should satisfy $\|\frac{\mathbf{X}\mathbf{X}^\mathsf{T}}{n} - \frac{\mathbf{Z}\mathbf{Z}^\mathsf{T}}{s}\|_2 \leq \frac{\sqrt{\varepsilon}}{\hat{c}^2}$.

Using the low rank approximation theorem stated in equation 13, we can say that $m$ needs to be at least the number of eigenvalues of $\frac{\mathbf{X}\mathbf{X}^\mathsf{T}}{n}$ larger than $\frac{\sqrt{\varepsilon}}{\hat{c}^2}$ or more formally the size of the synthetic dataset $m \geq \#\{\lambda_i(\mathbf{X}\mathbf{X}^\mathsf{T} > \frac{\sqrt{\varepsilon}}{\hat{c}^2}\} = \#\{\sigma_i(\mathbf{X}) > \frac{\varepsilon^{1/4}}{\hat{c}}\}$, where $\#$ denotes the size of a set and $\lambda, \sigma$ denote the eigenvalues and singular values of a matrix.

In the worst case choice of $D_{\text{train}}^{\text{sup}}$ (corresponding to $\mathbf{X} \in \mathbb{R}^{q \times n}$), we get a lower bound of $m \geq q = d+1$ on the size of $D_{\text{syn}}^{\text{sup}}$ when all $d+1$ singular values of $\zeta(D_{\text{train}}^{\text{sup}})$ are larger than $\frac{\varepsilon^{1/4}}{2}$.

## D.2 PROOF OF THEOREM 4.2

We begin with the following lemma.

**Lemma D.3.** *Let $d \geq 1$, $q = d+1$. For any $\mathbf{v}_1, \ldots, \mathbf{v}_T \in \mathbb{R}^q$ with $T < \frac{q(q+1)}{2}$ there exists a non-zero symmetric matrix $\mathbf{A} \in \mathbb{R}^{q \times q}$ that satisfies*

$$\mathbf{v}_t^\mathsf{T} \mathbf{A} \mathbf{v}_t = 0 \qquad for \ t = 1, \ldots, T. \tag{27}$$

*Proof.* Note that $\mathbf{v}_t^\mathsf{T} \mathbf{A} \mathbf{v}_t = \langle \mathbf{A}, \mathbf{v}_t \mathbf{v}_t^\mathsf{T} \rangle$. Further, the set of $q \times q$ symmetric matrices is a $q(q+1)/2$ dimensional linear subspace of $\mathbb{R}^{q \times q}$. Since $T < q(q+1)/2$, there exists a non-zero symmetric matrix in the orthogonal complement of the linear span of $\{\mathbf{v}_t \mathbf{v}_t^\mathsf{T}\}_{t=1}^T$ which can be taken to be the required matrix $\mathbf{A}$. $\qquad \square$

To complete the proof of Theorem 4.2 we first choose $D_{\text{train}}^{\text{sup}} = \{\mathbf{e}_1, \ldots, \mathbf{e}_q\}$ where $\mathbf{e}_i$ is the vector with 1 in the $i$th coordinate and zero otherwise i.e., it is the $i$th coordinate basis vector. It is easy to see that

$$L_{\text{mse}}(D_{\text{train}}^{\text{sup}}, f) = (1/q)\mathbf{v}^\mathsf{T}\mathbf{I}\mathbf{v} = (1/q)\|\mathbf{v}\|_2^2, \tag{28}$$

when $f(\mathbf{z}) := \mathbf{v}^\mathsf{T}\mathbf{z}$ for all $\mathbf{v} \in \mathbb{R}^q$. Now, let $f_1, \ldots, f_T$ be the regressors as in the statement of Theorem 4.2. Applying Lemma D.3 we obtain the symmetric matrix $\mathbf{A}$ which, by scaling, can be assumed to have largest magnitude eigenvalue i.e., its operator norm be $1/2$.

We now construct $D_{\text{syn}}^{\text{sup}}$ as follows. Consider the $\mathbf{B} = (\mathbf{I} + \mathbf{A})$. Since the operator norm of $\mathbf{A}$ is $1/2$, $\mathbf{B}$ is psd with maximum eigenvalue at most $3/2$ and minimum eigenvalue at least $1/2$. The eigen-decomposition of $\mathbf{B}$ implies that

$$\mathbf{B} = \sum_{i=1}^{q} (\sqrt{\lambda_i}\mathbf{u}_i)(\sqrt{\lambda_i}\mathbf{u}_i)^\mathsf{T} \tag{29}$$

where $\max_{i=1,\ldots,q} \lambda_i \leq 3/2$, $\min_{i=1,\ldots,q} \lambda_i \geq 1/2$, and $\|\mathbf{u}_i\|_2 = 1$ for $i = 1, \ldots, d$. We take $D_{\text{syn}}^{\text{sup}} := \{\sqrt{\lambda_i}\mathbf{u}_i\}_{i=1}^d$, so that its points have Euclidean norm at most $\sqrt{3/2} \leq 2$. Using equation 27 we have that,

$$L_{\text{mse}}(D_{\text{syn}}^{\text{sup}}, f_t) = (1/q)\mathbf{v}^\mathsf{T}\mathbf{B}\mathbf{v}_t = (1/q)(\|\mathbf{v}_t\|_2^2 + \mathbf{v}^\mathsf{T}\mathbf{A}\mathbf{v}_t) = (1/q)\|\mathbf{v}_t\|_2^2, \quad t = 1, \ldots, T. \tag{30}$$

The first condition in equation 6 directly follows from equation 28 and equation 30. To see the second condition, choose $\mathbf{v}_0$ to be the eigenvector of $\mathbf{A}$ corresponding to its maximum magnitude eigenvalue which is $1/2$. Then,

$$L_{\text{mse}}(D_{\text{syn}}^{\text{sup}}, f_0) = (1/q)(\|\mathbf{v}_0\|_2^2 + \mathbf{v}_0^\mathsf{T}\mathbf{A}\mathbf{v}_0) \tag{31}$$

which along with equation 28 implies that

$$L_{\text{mse}}^{\text{err}}(D_{\text{train}}^{\text{sup}}, D_{\text{syn}}^{\text{sup}}, f_0) = \left((1/q)\|\mathbf{v}_0\|_2^2 - (1/q)(\|\mathbf{v}_0\|_2^2 + \mathbf{v}_0^\mathsf{T}\mathbf{A}\mathbf{v}_0)\right)^2 = (1/q^2)(\mathbf{v}_0^\mathsf{T}\mathbf{A}\mathbf{v}_0)^2 \geq 1/(4q^2).$$

proving the second condition of equation 6 as well.

## E PROOF OF THEOREM 4.3

For ease of notation, we will use $O()$ notation to absorb absolute constants in the proof below.

For convenience, like in the case supervised regression case, we shall homogenize the Bellman loss as follows. Let us define $\zeta(s, a, r) := (t_1, \ldots, t_d, r)$ where $\phi(s, a) = (t_1, \ldots, t_d)$. Note that

since $\phi : \mathcal{B}(d_0, B_0) \times \mathcal{B}(d_0, B_0) \to \mathcal{B}(d, B)$, we have $\zeta : \mathcal{B}(d_0, B_0) \times \mathcal{B}(d_0, B_0) \times [0, R_{\max}] \to \mathcal{B}(q, B + R_{\max})$. We also define a class of functions $\mathcal{Q}$ mapping $\mathcal{B}(d_0, B_0) \times \mathcal{B}(d_0, B_0) \times [0, R_{\max}]$ to $\mathbb{R}$ where each $h \in \mathcal{Q}$ is given by $h(s, a, r) := \mathbf{r}^\mathsf{T} \zeta(s, a, r)$ for some $\mathbf{r} \in \mathbb{R}^{d+1}$. Let $\mathcal{Q}_1$ be the restricted class where $\|\mathbf{r}\|_2 \le 2$

Note that for any $f \in \mathcal{Q}_0$ given by $f(s, a) := \mathbf{v}^\mathsf{T} \phi(s, a)$, taking $\mathbf{r} = (v_1, \ldots, v_d, -\lambda)$ yields that $h \in \mathcal{Q}$ and $h(s, a, r) = f(s, a) - \lambda r$, and $f(s', a') = h(s', a', 0)$. Thus, we define another version of the Bellman loss as:

$$\tilde{L}_{\text{Bell}}(D, h) := \mathbb{E}_{(s,a,r,s') \leftarrow D} \left[ \left( h(s, a, r) - \gamma \max_{a' \in \mathcal{A}} h(s', a', 0) \right)^2 \right] \tag{32}$$

Using this we can see that, for $f \in \mathcal{Q}_0$ where $f(s, a) := \mathbf{v}^\mathsf{T} \phi(s, a)$, taking $\mathbf{r} = (v_1, \ldots, v_d, -1)$ and $h(s, a, r) := \mathbf{r}^\mathsf{T} \zeta(s, a, r)$ we obtain that $h \in \mathcal{Q}_1$ and

$$L_{\text{Bell}}(D, f) = \tilde{L}_{\text{Bell}}(D, h). \tag{33}$$

Further, for some $(f, \lambda) \in H$, where $f(s, a) := \mathbf{v}^\mathsf{T} \phi(s, a)$, letting $\mathbf{r} = (v_1, \ldots, v_d, -\lambda)$ and $h(s, a, r) := \mathbf{r}^\mathsf{T} \zeta(s, a, r)$, we obtain that $h \in \mathcal{Q}$ and

$$\hat{L}_{\text{Bell}}(D, f, \lambda) = \tilde{L}_{\text{Bell}}(D, h). \tag{34}$$

Thus, by abuse of notation, we think of $h \in \mathcal{H}$ being chosen randomly by sampling $\mathbf{r} \in N(0,1)^{d+1}$. We shall use $q$ to denote $d + 1$ in the rest of this section. We fix $D_{\text{train}}^{\text{orl}} = \{(s_i, a_i, r_i, s_i')\}_{i=1}^n$ as the given training dataset, and for $\hat{D} = \{(\hat{s}_i, \hat{a}_i, \hat{r}_i, \hat{s}_i') \in \mathcal{B}(d_0, B_0) \times \mathcal{B}(d_0, B_0) \times [0, R_{\max}] \times \mathcal{B}(d_0, B_0)\}_{i=1}^t$, we define:

$$\tilde{L}_{\text{Bell}}^{\text{err}}(D_{\text{train}}^{\text{orl}}, \hat{D}, h) := \left( \tilde{L}_{\text{Bell}}(D_{\text{train}}^{\text{orl}}, h) - \tilde{L}_{\text{Bell}}(\hat{D}, h) \right)^2 \tag{35}$$

It is easy to see using the norm bounds and equation 32, equation 35 that for any $h \in \mathcal{Q}_1$, $\tilde{L}_{\text{Bell}}(D, h), \tilde{L}_{\text{Bell}}(D_{\text{train}}^{\text{orl}}, h) \le 16(B + R_{\max})^2$ and $\tilde{L}_{\text{Bell}}^{\text{err}}(D_{\text{train}}^{\text{orl}}, \hat{D}, h) \le 256(B + R_{\max})^4$.

### E.1 Probabilistic Lower bound for fixed $\hat{h}$ and $\hat{D}$

We fix for this subsection $\hat{h} \in \mathcal{Q}_1$ and $\hat{D}$ as above. First we prove the following lemma.

**Lemma E.1.** *If $\tilde{L}_{\text{Bell}}^{\text{err}}(D_{\text{train}}^{\text{orl}}, \hat{D}, \hat{h}) \ge \Delta$, then over the choice of $g \in H$, s.t. $g(s, a, r) := \mathbf{r}^\mathsf{T} \zeta(s, a, r)$,*

$$\Pr\left[ \left( \tilde{L}_{\text{Bell}}^{\text{err}}(D_{\text{train}}^{\text{orl}}, \hat{D}, g) \ge \Delta/8 \right) \wedge (\|\mathbf{r}\|_2 \le 2) \right] \ge \left( \frac{\Delta}{(\sqrt{d}(B + R_{\max})^4))} \right)^{(c_1 d)}. \tag{36}$$

*where $c_1 > 0$ is a constant.*

*Proof.* Let $\hat{h}$ correspond to the vector $\hat{c}\hat{\mathbf{r}} \in \mathbb{R}^{d+1}$ where $\|\hat{\mathbf{r}}\|_2 = 1$ and $\hat{c} \le 2$. Thus, a random $\mathbf{r} \in N(0,1)^{d+1}$ corresponds to $\alpha \hat{\mathbf{r}} + \mathbf{H}\tilde{\mathbf{r}}$ where $\alpha \sim N(0,1)$ and $\tilde{\mathbf{r}} \sim N(0,1)^d$ are independently sampled and $\mathbf{H}$ is a $(d+1) \times d$ matrix of unit norm columns which are a completion of $\hat{\mathbf{r}}$ to a orthonormal basis. From the above upper bound, we have that $\tilde{L}_{\text{Bell}}^{\text{err}}(D_{\text{train}}^{\text{orl}}, \hat{D}, \hat{h}) \le 256(B + R_{\max})^4$ and therefore, $\Delta \le 256(B + R_{\max})^4$.

Let us condition on the event $\mathcal{E}$ that $\|\tilde{\mathbf{r}}\|_\infty \le \left( c_0 \Delta / (\sqrt{d}(B + R_{\max})^4)) \right)$ and $\alpha \in [1, 3/2]$. Firstly, since $\left( c_0 \Delta / (\sqrt{d}(B + R_{\max})^4)) \right) = O(1)$, by the properties of the standard Gaussian distribution, one obtains that

$$\Pr[\mathcal{E}] \ge \left( \frac{\Delta}{\sqrt{d}(B + R_{\max})^4} \right)^{c_1 d} \tag{37}$$

for some $c_1$ depending on $c_0$ only. Now, $\mathcal{E}$ implies that $\|\mathbf{H}\tilde{\mathbf{r}}\|_2 \leq \left(c_0\Delta/(B+R_{\max})^4\right) = \upsilon < 0.1$ for which we choose $c_0 > 0$ small enough. Since $\alpha > 0$ under $\mathcal{E}$ this implies that

$$
\begin{aligned}
&\left( g(s,a,r) - \gamma \max_{a'\in\mathcal{A}} g(s',a',0) \right)^2 \\
&= \left( (\alpha\hat{\mathbf{r}} + \mathbf{H}\tilde{\mathbf{r}})^\mathsf{T}\zeta(s,a,r) - \gamma \max_{a'\in\mathcal{A}} (\alpha\hat{\mathbf{r}} + \mathbf{H}\tilde{\mathbf{r}})^\mathsf{T}\zeta(s',a',0) \right)^2 \\
&= \left( \alpha\hat{\mathbf{r}}^\mathsf{T}\zeta(s,a,r) - \gamma \max_{a'\in\mathcal{A}} \alpha\hat{\mathbf{r}}^\mathsf{T}\zeta(s',a',0) \pm O(\upsilon(B+R_{\max})) \right)^2 \\
&= \left( \alpha\hat{\mathbf{r}}^\mathsf{T}\zeta(s,a,r) - \alpha\gamma \max_{a'\in\mathcal{A}} \hat{\mathbf{r}}^\mathsf{T}\zeta(s',a',0) \pm O(\upsilon(B+R_{\max})) \right)^2 \\
&= \left( \alpha\hat{\mathbf{r}}^\mathsf{T}\zeta(s,a,r) - \alpha\gamma \max_{a'\in\mathcal{A}} \hat{\mathbf{r}}^\mathsf{T}\zeta(s',a',0) \right)^2 \pm O(\upsilon(B+R_{\max})^2)
\end{aligned}
\tag{38}
$$

where we used the upper bound on $\alpha$. Thus, using analysis similar to the proof of Lemma C.3, we obtain that for some choice of $c_0 > 0$ small enough, $\mathcal{E}$ implies that

$$
\left| \tilde{L}_{\mathrm{Bell}}^{\mathrm{err}}(D_{\mathrm{train}}^{\mathrm{orl}}, \hat{D}, g) - \alpha^4 \tilde{L}_{\mathrm{Bell}}^{\mathrm{err}}(D_{\mathrm{train}}^{\mathrm{orl}}, \hat{D}, (\hat{h}/\hat{c})) \right| \leq \Delta/64
\tag{39}
$$

and since $\alpha \geq 1$ and and $\hat{c} \leq 2$, we have $\alpha^4 \tilde{L}_{\mathrm{Bell}}^{\mathrm{err}}(D_{\mathrm{train}}^{\mathrm{orl}}, \hat{D}, (\hat{h}/\hat{c})) \geq \Delta/16$. Thus,

$$
\tilde{L}_{\mathrm{Bell}}^{\mathrm{err}}(D_{\mathrm{train}}^{\mathrm{orl}}, \hat{D}, g) \geq \Delta/32.
\tag{40}
$$

Moreover, since $\|\mathbf{H}\tilde{\mathbf{r}}\|_2 < 0.1$ and $\alpha \in [1, 3/2]$, we obtain that $\|\mathbf{r}\|_2 \leq 2$, which completes the proof. $\qquad\square$

The following amplified version of the previous lemma is a direct implication.

**Lemma E.2.** *Let* $\tilde{L}_{\mathrm{Bell}}^{\mathrm{err}}(D_{\mathrm{train}}^{\mathrm{orl}}, \hat{D}, \hat{h}) \geq \Delta$. *Then, for iid random* $g_1, \ldots, g_k \sim H$, *s.t.* $g_j(s,a,r) := \mathbf{r}_j^\mathsf{T}\zeta(s,a,r)$ *we have*

$$
\Pr\left[ \bigvee_{j=1}^{k} \left( \left( \tilde{L}_{\mathrm{Bell}}^{\mathrm{err}}(D_{\mathrm{train}}^{\mathrm{orl}}, \hat{D}, g_j) \geq \Delta/8 \right) \wedge (\|\mathbf{r}_j\|_2 \leq 2) \right) \right] \geq 1 - \left( 1 - \left( \frac{\Delta}{(\sqrt{d}(B+R_{\max})^4)} \right)^{(c_1 d)} \right)^k
$$

## E.2 NET OVER PREDICTORS $h$ AND SYNTHETIC DATASETS $D_{\mathrm{syn}}^{\mathrm{orl}}$

First we shall construct a net over predictors $h \in \mathcal{Q}_1$, where $h(\mathbf{x}) := \mathbf{r}^\mathsf{T}\zeta(s,a,r)$ for some $\mathbf{r} \in \mathcal{B}(q, 2)$. Thus, we can consider the Euclidean net $\mathcal{T}(q, 2, \xi))$ and let $\hat{\mathcal{Q}}_1 := \{\hat{h} \in \mathcal{Q}_1 : \exists \hat{\mathbf{r}} \in \mathcal{T}(q, 2, \xi) \text{ s.t. } \hat{h}(s,a,r) := \hat{\mathbf{r}}^\mathsf{T}\zeta(s,a,r)\}$, for some parameter $\xi \in (0,1)$ to be chosen later. The proof of the following simple approximation lemma is exactly along the lines of the proof of Lemma C.3 and the analysis in the previous subsection accounting for the error in the max term of equation 32 which is handled similarly.

**Lemma E.3.** *For any* $D_{\mathrm{train}}^{\mathrm{orl}}$ *and* $\hat{D}$ *as defined in the previous subsection, for any* $h \in \mathcal{Q}_1$, $\exists \hat{h} \in \hat{\mathcal{Q}}_1$ *s.t.*

$$
\left| \tilde{L}_{\mathrm{Bell}}^{\mathrm{err}}(D_{\mathrm{train}}^{\mathrm{orl}}, \hat{D}, h) - \tilde{L}_{\mathrm{Bell}}^{\mathrm{err}}(D_{\mathrm{train}}^{\mathrm{orl}}, \hat{D}, \hat{h}) \right| \leq 300\xi(B+R_{\max})^4.
\tag{41}
$$

The argument for net over the synthetic dataset is similar - we show that the synthetic data $D_{\mathrm{syn}}^{\mathrm{orl}}$ can be approximated with a much smaller dataset $\hat{D}$ whose points are from a discrete set.

**Lemma E.4.** *Fix a* $\tau, \nu \in (0,1)$. *For any* $D_{\mathrm{syn}}^{\mathrm{orl}}$, *there exists a dataset* $\hat{D}$ *of size* $s := O((p/\tau^2)\log(1/\tau))$ *(where* $p$ *is the pseudo-dimension in Sec. 4) whose points* $(\hat{s}, \hat{a}, \hat{r}, \hat{s}')$ *are s.t.* $\hat{s}, \hat{a}, \hat{s}'$ *are from the Euclidean net* $\mathcal{T}(d_0, B_0, \nu(B+R_{\max})/(10L))$ *and* $\hat{r} \in \mathcal{T}(1, R_{\max}, \nu(B+R_{\max}))$ *such that for any* $h \in \mathcal{Q}_1$,

$$
\left| \tilde{L}_{\mathrm{Bell}}^{\mathrm{err}}(D_{\mathrm{train}}^{\mathrm{orl}}, \hat{D}, h) - \tilde{L}_{\mathrm{Bell}}^{\mathrm{err}}(D_{\mathrm{train}}^{\mathrm{orl}}, D_{\mathrm{syn}}^{\mathrm{orl}}, h) \right| \leq O(\tau(B+R_{\max})^4 + \nu(B+R_{\max})^4)
\tag{42}
$$

*Proof.* Since $p$ is the pseudo-dimension is as defined in Sec. 4), it is also the pseudo-dimension on the class of functions $\mathcal{V}$ given by $\tilde{L}_{\text{Bell}}(\{s, a, r, s'\}, h)$ for $h \in \mathcal{Q}_1$. Thus, Let us first define $\overline{D}$ probabilistically to be a set of $t := O((p/\tau^2) \log(1/\tau))$ points independently and u.a.r. sampled from $D_{\text{syn}}^{\text{orl}}$. Using the upper bound on $\mathcal{N}(\tau/16, \mathcal{V}, 2t)$ from Chapters 10.4 and 12.3, and Theorem 17.1 of [Anthony-Bartlett, 2009], we get that for any $h \in \mathcal{Q}_1$,

$$\left| \tilde{L}_{\text{Bell}}(\overline{D}, h) - \tilde{L}_{\text{Bell}}(D_{\text{syn}}^{\text{orl}}, h) \right| \leq 16\tau(B + R_{\max})^2 \tag{43}$$

with probability at least $> 0$. Thus, using the above and arguments similar to those used earlier in this and the previous sections we obtain that there exists $\overline{D}$ of size $s$ s.t.

$$\left| \tilde{L}_{\text{Bell}}^{\text{err}}(D_{\text{train}}^{\text{orl}}, \overline{D}, h) - \tilde{L}_{\text{Bell}}^{\text{err}}(D_{\text{train}}^{\text{orl}}, D_{\text{syn}}^{\text{orl}}, h) \right| \leq O(\tau(B + R_{\max})^4) \tag{44}$$

Lastly, to get $\hat{D}$ we replace each $(s, a, r, s')$ in $\overline{D}$ with the nearest point in the net $\mathcal{T}_1 := \mathcal{T}(d_0, B_0, \nu(B + R_{\max})/(10L)) \times \mathcal{T}(d_0, B_0, \nu(B + R_{\max})/(10L)) \times \mathcal{T}(1, R_{\max}, \nu(B + R_{\max}))) \times \mathcal{T}(d_0, B_0, \nu(B + R_{\max})/(10L))$. Using the Lipschitzness bound of $L$ on $\phi$ and the analysis used in the proof of Lemma C.4, we obtain that,

$$\left| \tilde{L}_{\text{Bell}}^{\text{err}}(D_{\text{train}}^{\text{orl}}, \overline{D}, h) - \tilde{L}_{\text{Bell}}^{\text{err}}(D_{\text{train}}^{\text{orl}}, \hat{D}, h) \right| \leq O(\nu(B + R_{\max})^4) \tag{45}$$

Combining the above two inequalities completes the proof. $\qquad\square$

### E.3 Union bound and proof of Theorem 4.3

We take $\xi, \tau$ and $\nu$ to be $O(\Delta/(B + R_{\max})^4)$, so that applying Lemmas E.4 and E.3 yields,

**Lemma E.5.** *For any $h \in \mathcal{Q}_1$ and $D_{\text{syn}}^{\text{orl}}$, there exist $\hat{h} \in \hat{\mathcal{Q}}_1$ and $\hat{D} \in (\mathcal{T}_1)^t$ such that*

$$\left| \tilde{L}_{\text{Bell}}^{\text{err}}(D_{\text{train}}^{\text{orl}}, D_{\text{syn}}^{\text{orl}}, h) - \tilde{L}_{\text{Bell}}^{\text{err}}(D_{\text{train}}^{\text{orl}}, \hat{D}, \hat{h}) \right| \leq C'\Delta \tag{46}$$

*where $C' > 0$ is a small enough constant and $t := p(C_1/\tau^2) \log(1/\tau)$ for some constant $C_1$ and $\hat{\mathcal{Q}}_1$ is as defined in the previous subsection.*

Now, the size of the net is bounded as follows:

$$\left| \hat{\mathcal{Q}}_1 \right| \times |\mathcal{T}_1|^t \leq \left( \frac{6}{\xi} \right)^q \cdot \left( \frac{6B_0 L}{\nu(B + R_{\max})} \right)^{d_0 t}$$
$$\leq \exp\left( O(\ell((B + R_{\max})^4/\Delta)^2) \right) \tag{47}$$

where

$$\ell := (q \log((B + R_{\max})/\Delta) + d_0 p \log(B_0 L(B + R_{\max})/\Delta) \tag{48}$$

Now, using the above, we take $k = \left( \frac{\Delta}{(\sqrt{d}(B + R_{\max})^4))} \right)^{(-O(d))} \ell \log(1/\delta)$ in Lemma E.2 and apply union bound over the net (along the lines of the proof in Sec. C.3). To get back the lower bound with $D_{\text{syn}}^{\text{orl}}$ we apply Lemma E.4 to $g_j$ such that $|\mathbf{r}_j| \leq 2$ i.e. $g_j \in \mathcal{Q}_1$. This completes the proof.

## F Proof of Theorem 4.4

We can use elementary manipulation of $L_{\text{Bell}}^{\text{err}}$ based on equation 11 and the additive state-action embedding to essentially remove the max term and the subsequent proof is along the same lines as that of Theorem 4.1.

We have $f(s, a) := \mathbf{v}^\top \phi(s, a) = \mathbf{v}^\top \phi_1(s) + \mathbf{v}^\top \phi_2(a)$ as defined, so notice that

$$\max_{a' \in \mathcal{A}} f(s', a') = \mathbf{v}^\top \phi_1(s') + \max_{a' \in \mathcal{A}} \mathbf{v}^\top \phi_2(a') \tag{49}$$

In equation 49, $\max_{a' \in \mathcal{A}} \mathbf{v}^\top \phi_2(a')$ is a constant independent of the training and synthetic dataset and depends only on $\mathbf{v}$, so we define $\alpha(\mathbf{v}) = \max_{a' \in \mathcal{A}} \mathbf{v}^\top \phi_2(a')$. Expanding within the outer square, canceling and zeroing out using mild linear conditions in equation 11 and 49, we obtain

$$
\begin{aligned}
& L_{\text{Bell}}^{\text{err}}(D_{\text{train}}^{\text{orl}}, D_{\text{syn}}^{\text{orl}}, f, \lambda) \\
={} & (\mathbb{E}_{(s,a,r,s') \leftarrow D_{\text{train}}^{\text{orl}}}[(f(s,a) - \lambda r - \gamma \max_{a' \in \mathcal{A}} f(s',a'))^2] \\
& - \mathbb{E}_{(s,a,r,s') \leftarrow D_{\text{syn}}^{\text{orl}}}[(f(s,a) - \lambda r - \gamma \max_{a' \in \mathcal{A}} f(s',a'))^2])^2 \\
={} & (\mathbb{E}_{(s,a,r,s') \leftarrow D_{\text{train}}^{\text{orl}}}[(\mathbf{v}^\top \phi_1(s) + \mathbf{v}^\top \phi_2(a) - \lambda r - \gamma \mathbf{v}^\top \phi_1(s') - \gamma \alpha(\mathbf{v}))^2] \\
& - \mathbb{E}_{(s,a,r,s') \leftarrow D_{\text{syn}}^{\text{orl}}}[(\mathbf{v}^\top \phi_1(s) + \mathbf{v}^\top \phi_2(a) - \lambda r - \gamma \mathbf{v}^\top \phi_1(s') - \gamma \alpha(\mathbf{v}))^2])^2 \\
={} & (\mathbb{E}_{(s,a,r,s') \leftarrow D_{\text{train}}^{\text{orl}}}[(\mathbf{v}^\top (\phi_1(s) - \gamma \phi_1(s') + \phi_2(a)) - \lambda r - \gamma \alpha(\mathbf{v}))^2] \\
& - \mathbb{E}_{(s,a,r,s') \leftarrow D_{\text{syn}}^{\text{orl}}}[(\mathbf{v}^\top (\phi_1(s) - \gamma \phi_1(s') + \phi_2(a)) - \lambda r - \gamma \alpha(\mathbf{v}))^2])^2 \\
={} & (\mathbb{E}_{(s,a,r,s') \leftarrow D_{\text{train}}^{\text{orl}}}[(\mathbf{v}^\top (\phi_1(s) - \gamma \phi_1(s') + \phi_2(a)) - \lambda r)^2] \\
& - \mathbb{E}_{(s,a,r,s') \leftarrow D_{\text{train}}^{\text{syn}}}[(\mathbf{v}^\top (\phi_1(s) - \gamma \phi_1(s') + \phi_2(a)) - \lambda r)^2] \\
& + \mathbb{E}_{(s,a,r,s') \leftarrow D_{\text{train}}^{\text{orl}}}[\gamma^2 \alpha^2(\mathbf{v})] - \mathbb{E}_{(s,a,r,s') \leftarrow D_{\text{train}}^{\text{syn}}}[\gamma^2 \alpha^2(\mathbf{v})] \\
& - 2\mathbb{E}_{(s,a,r,s') \leftarrow D_{\text{train}}^{\text{orl}}}[\gamma \alpha(\mathbf{v})(\mathbf{v}^\top (\phi_1(s) - \gamma \phi_1(s') + \phi_2(a)) - \lambda r)] \\
& + 2\mathbb{E}_{(s,a,r,s') \leftarrow D_{\text{train}}^{\text{syn}}}[\gamma \alpha(\mathbf{v})(\mathbf{v}^\top (\phi_1(s) - \gamma \phi_1(s') + \phi_2(a)) - \lambda r)])^2 \\
={} & (\mathbb{E}_{(s,a,r,s') \leftarrow D_{\text{train}}^{\text{orl}}}[(\mathbf{v}^\top (\phi_1(s) - \gamma \phi_1(s') + \phi_2(a)) - \lambda r)^2] \\
& - \mathbb{E}_{(s,a,r,s') \leftarrow D_{\text{syn}}^{\text{orl}}}[(\mathbf{v}^\top (\phi_1(s) - \gamma \phi_1(s') + \phi_2(a)) - \lambda r)^2])^2 \quad (50)
\end{aligned}
$$

Now notice that the loss term equation 50 looks exactly like the linear supervised regression loss $L_{\text{mse}}^{\text{err}}$. We can concatenate $\mathbf{v}$ and $\lambda$ to get a regression vector $\mathbf{r}$. In addition, we take the feature vectors $\mathbf{x}$ to be simply $\phi_1(s) - \gamma \phi_1(s') + \phi_2(a)$ and the regression label $y$ to be $r$. Note that $\mathbf{x} \leq O(B)$ and $|y| \leq R_{\max}$.

We can sample regressors $\mathbf{r} \sim N(0, 1/(d+1))^{d+1}$ u.a.r. and use the concatenation trick as before. The nets will be taken over the embeddings of states and actions instead of the states and actions themselves. Thus, via analysis similar to the linear regression case (Theorem 4.1), we obtain a $O\left(d^2 \log(d(B + R_{\max})/\Delta) \log(1/\delta)\right)$ upper bound on the number of sampled predictors, given that the synthetic dataset satisfies the mild linear condition above.

## F.1 CONVEXITY OF THE OBJECTIVE

As seen above, $\hat{L}_{\text{Bell}}^{\text{err}}$ reduces to the supervised regression case with $\phi_1(s) - \gamma \phi_1(s') + \phi_2(a)$ as the regression point with regression label $r$, corresponding to $(s, a, r, s')$. From Appendix G.2, the optimization objective in the supervised regression case is convex in the synthetic data regression points and labels.

Therefore, if $\phi_1$ and $\phi_2$ are linear in $(s, a, r, s')$ then the $\hat{L}_{\text{Bell}}^{\text{err}}$ is also convex in the points of $D_{\text{syn}}^{\text{orl}}$. Further, the condition in equation 11 is also an affine linear constraint on the points of $D_{\text{syn}}^{\text{orl}}$, and therefore the optimization is convex.

## G USEFUL TECHNICAL DETAILS

### G.1 BELLMAN LOSS RISK BOUNDS

The *value gap* i.e., the difference between the true and estimated value functions for a policy corresponding to a $Q$-value predictor can be estimated by the latter's empirical Bellman Loss on offline data. This holds under under a certain *concentrability* assumption – informally it states that the dataset should contain all state-action pairs which are reachable from the starting state with significant probability, should also be present with sufficient probability in the dataset. We refer the reader to Assumption 1 and Lemma 3.2 of (Duan et al., 2021) for more details.

Table 1: Homogeneous Linear Regression MSE loss over *white and red wine quality* data

| wine color | $N_{\mathrm{syn}}$ | $D_{\mathrm{train}}$ | $D_{\mathrm{syn}}$ | $D_{\mathrm{rand}}$ | $D_{\mathrm{lev}}$ |
|---|---|---|---|---|---|
| *red* | 20 | | $0.85 \pm 0.20$ | $1.82 \pm 1.22$ | $1.21 \pm 0.45$ |
| | 50 | $0.65 \pm 0.04$ | $0.65 \pm 0.07$ | $0.97 \pm 0.22$ | $0.80 \pm 0.08$ |
| | 100 | | $0.67 \pm 0.05$ | $0.81 \pm 0.16$ | $0.73 \pm 0.07$ |
| *white* | 20 | | $1.00 \pm 0.21$ | $1.81 \pm 0.44$ | $1.31 \pm 0.30$ |
| | 50 | $0.74 \pm 0.04$ | $0.82 \pm 0.17$ | $1.04 \pm 0.19$ | $0.90 \pm 0.11$ |
| | 100 | | $0.75 \pm 0.02$ | $0.83 \pm 0.05$ | $0.84 \pm 0.05$ |

## G.2 CONVEXITY OF $L_{\mathrm{mse}}^{\mathrm{err}}(D_{\mathrm{train}}^{\mathrm{sup}}, D_{\mathrm{syn}}^{\mathrm{sup}}, f)$

From equation 1, $L_{\mathrm{mse}}(D_{\mathrm{train}}^{\mathrm{sup}}, f)$ is constant w.r.t. $D_{\mathrm{syn}}^{\mathrm{sup}}$ and the dependent part is $L_{\mathrm{mse}}(D_{\mathrm{syn}}^{\mathrm{sup}}, f)$. The latter is the mean squared error loss which is the average of squares of linear functions of the points of $D_{\mathrm{syn}}^{\mathrm{sup}}$ and is therefore convex in the points of $D_{\mathrm{syn}}^{\mathrm{sup}}$. Since $L_{\mathrm{mse}}^{\mathrm{err}}(D_{\mathrm{train}}^{\mathrm{sup}}, D_{\mathrm{syn}}^{\mathrm{sup}}, f)$ is convex in $L_{\mathrm{mse}}(D_{\mathrm{syn}}^{\mathrm{sup}}, f)$, it is also convex in the points of $D_{\mathrm{syn}}^{\mathrm{sup}}$.

Now, if $\phi$ is linear in $(s, a)$ (the concatenation of $s$ and $a$), then $\hat{L}_{\mathrm{Bell}}(D_{\mathrm{syn}}^{\mathrm{orl}}, f, \lambda)$ (from equation 7) is a sum of squares of terms which are each the difference of (i) a linear function in the points of $D_{\mathrm{syn}}^{\mathrm{orl}}$ and, (ii) the maximum over a $\mathcal{A}$ of linear functions of the points of $D_{\mathrm{syn}}^{\mathrm{orl}}$. Since max (over a fixed number of arguments) is a convex function over its vector of arguments, this implies that $\hat{L}_{\mathrm{Bell}}(D_{\mathrm{syn}}^{\mathrm{orl}}, f, \lambda)$ is convex and therefore $\hat{L}_{\mathrm{Bell}}^{\mathrm{err}}(D_{\mathrm{train}}^{\mathrm{orl}}, D_{\mathrm{syn}}^{\mathrm{orl}}, f, \lambda)$ is also convex.

## H ADDITIONAL EXPERIMENTS

### H.1 ADDITIONAL EXPERIMENTAL DETAILS FOR THE WINE QUALITY DATASET

The red wine dataset has 1599 wine samples and the white wine dataset has 4898 wine samples. For both red and white wines, we use the feature QUALITY as the label and regress on the remaining 11 features. We pre-process the data by standardizing each feature column and label. We randomly shuffle the samples with an 80/20 split into training and test data.

We use a linear homogeneous model (i.e. $f(\mathbf{x}) = \mathbf{r}^{\mathsf{T}}\mathbf{x}$) for regression on the label. We sample $k = 100$ linear regressors from the standard normal distribution as well as $N_{\mathrm{eval}} = 100$ another regressors from the same distribution. Using a fixed learning rate of $0.01$ on the Adam optimizer and at most 5000 steps, we optimize the randomly initialized synthetic data until the objective has minimum value on the $N_{\mathrm{eval}} = 100$ regressors. The model is then trained on the four datasets using the Adam optimizer with a learning rate of $0.001$ and the mean loss is reported in Table 1.

We observe that a model trained on $D_{\mathrm{syn}}$ performs better than ones trained on both $D_{\mathrm{rand}}$ and $D_{\mathrm{lev}}$. We also observe that the test loss of a model trained on $D_{\mathrm{syn}}$ decreases, as expected.

### H.2 ADDITIONAL EXPERIMENTAL DETAILS FOR THE BOSTON HOUSING DATASET

The Boston Housing dataset has 506 samples. We use the feature MEDV (Median value of owner-occupied homes in 1000's) as the label and regress on the remaining 13 features. We preprocess the data by standardizing each feature column and label. We randomly shuffle the samples with an 80/20 split into training and test data. The sizes of the synthetic dataset $N_{\mathrm{syn}}$ we consider are $\{20, 50, 100\}$.

We take $k = 100$ random homogeneous linear regressors in Algorithm 1. Using a fixed learning rate of $0.01$ on the Adam optimiser and at most 5000 steps, we optimise the randomly initialised synthetic data until the objective has minimum value on the $N_{\mathrm{eval}} = 100$ randomly sampled regressors. The model is then trained on the four datasets using the Adam optimiser with a learning rate of $0.001$. The mean test loss of models trained on respective datasets (over 10 trials) are included in Table 2.

We observe that a model trained on $D_{\mathrm{syn}}$ performs better than one trained on $D_{\mathrm{rand}}$ and on par with $D_{\mathrm{lev}}$. We also observe that the test loss of a model trained on $D_{\mathrm{syn}}$ decreases, as expected.

Table 2: Homogeneous Linear Regression MSE loss over *Boston Housing* data

| $N_{\text{syn}}$ | $D_{\text{train}}$ | $D_{\text{syn}}$ | $D_{\text{rand}}$ | $D_{\text{lev}}$ |
|---|---|---|---|---|
| 20 | | $0.57 \pm 0.23$ | $1.77 \pm 1.71$ | $0.57 \pm 0.20$ |
| 50 | $0.27 \pm 0.05$ | $0.40 \pm 0.14$ | $0.44 \pm 0.16$ | $0.38 \pm 0.08$ |
| 100 | | $0.29 \pm 0.07$ | $0.29 \pm 0.08$ | $0.36 \pm 0.09$ |

Table 3: Homogeneous Linear Regression MSE loss over *California Housing* data

| $N_{\text{syn}}$ | $D_{\text{train}}$ | $D_{\text{syn}}$ | $D_{\text{rand}}$ |
|---|---|---|---|
| 10 | | $1.11 \pm 0.27$ | $0.99 \pm 0.05$ |
| 20 | | $1.13 \pm 0.33$ | $0.94 \pm 0.07$ |
| 50 | $0.56 \pm 0.05$ | $1.16 \pm 0.30$ | $0.73 \pm 0.06$ |
| 100 | | $1.16 \pm 0.71$ | $0.73 \pm 0.07$ |
| 200 | | $0.94 \pm 0.38$ | $0.69 \pm 0.02$ |

### H.3 ADDITIONAL EXPERIMENTS ON THE CALIFORNIA HOUSING DATASET

For the supervised case we propose to add experiments on the California Housing dataset (Pace & Barry, 1997) which has around $20K$ points and 9 features. The dataset distillation is done using linear models, while the training and evaluation is done using a $(10, 10)$ two hidden layer ReLU based neural network, demonstrating that our synthetic data generation process can in practice be used with different model architectures for evaluation. We sample $k = 100$ random linear models for distillation.

We see from Table 3 that our distillation technique significantly outperforms random sampling and achieves 200-factor compression, even on large datasets like California Housing.

### H.4 ADDITIONAL EXPERIMENTS AND DETAILS FOR THE CARTPOLE ENVIRONMENT

In this environment, a pole is attached by an un-actuated joint to a cart, which moves along a frictionless track. The pendulum is placed upright on the cart and the goal is to balance the pole by applying forces in the left and right direction on the cart. The action indicates the direction of the fixed force the cart is pushed with and can take two discrete values. The observation is an array with shape $(4, )$ with the values corresponding to the positions and velocities. The episode terminates if the pole angle or position goes beyond a certain range and is artificially truncated after 500 time-steps. Since the goal is to keep the pole upright for as long as possible, by default, a reward of $+1$ is given for every step taken, including the termination step.

We sample $n = 10000$ steps of completely random policies in this environment to get $D_{\text{train}}$, which contains both terminated and non-terminated states. We separate them into $D_{\text{train–terminated}}$ and $D_{\text{train-nonterminated}}$. We have to do this in our implementation as the theoretical result only analyzes infinite horizon MDP's. In practice, the $q$-value function for the terminated states does not have any max term as the trajectory ends there, i.e. for a terminated state $s_{\text{term}}$, the q-value is simply the reward, so we use $f(s_{\text{term}}, a) = r$. We use our data distillation routine to get a distilled version of each of these datasets separately and combine them together to get $D_{\text{syn-nonterminated}} + D_{\text{syn-terminated}} = D_{\text{syn}}$. Note that the ratio of the terminated and non-terminated states in $D_{\text{syn}}$ and $D_{\text{train}}$ is artificially kept the same and we take the total size of the synthetic dataset $D_{\text{syn}}$ to be $N_{\text{syn}}$.

Our model architecture for this experiment is fixed to a 2-layer (10,10) ReLU based neural network. For the synthetic data generation, we sample $k$ models and we use the Adam optimiser and perform a search over the learning rate among {3e-1, 1e-1, 3e-2, 1e-2, 3e-3, 1e-3, 3e-4, 1e-4} for the distillation processes of both the terminated and non-terminated states and report the synthetic dataset that performs the best on evaluation.

We train the three datasets using the Fitted-Q iteration algorithm that iteratively optimises the Bellman loss to get a $q$-value predictor. We test the optimal policy corresponding to the $q$-value predictor on the real environment and report the returns. We report an extensive evaluation with multiple values of $k, N_{\text{syn}}$ in Table 4. We observe that on average, policies trained on our generated synthetic data perform better than both real and random data. As expected, we see that as the size of $D_{\text{syn}}$

increases (i.e. $N_{syn}$ increases), the model trained on it performs better. Surprisingly, increasing the number of randomly sampled action-value predictors $k$ does not seem to have a discernible impact on performance.

Table 4: Extended Evaluation for the Cartpole Environment with $(10, 10)$-layer NN.

| $k$ | $N_{syn}$ | $D_{train}$ | $D_{rand}$ | $D_{syn}$ |
|---|---|---|---|---|
| 5 | 10 | $339.10 \pm 132.07$ | $13.50 \pm 0.67$ | $500.00 \pm 0.00$ |
| 10 | 10 | $332.70 \pm 129.18$ | $13.90 \pm 0.54$ | $500.00 \pm 0.00$ |
| 20 | 10 | $416.30 \pm 133.73$ | $13.50 \pm 0.50$ | $455.40 \pm 133.80$ |
| 50 | 10 | $343.10 \pm 129.38$ | $13.40 \pm 0.66$ | $181.00 \pm 21.87$ |
| 100 | 10 | $368.20 \pm 116.77$ | $13.30 \pm 1.19$ | $464.00 \pm 108.00$ |
| 5 | 20 | $332.10 \pm 117.26$ | $135.20 \pm 41.38$ | $500.00 \pm 0.00$ |
| 10 | 20 | $377.20 \pm 122.43$ | $148.20 \pm 43.15$ | $240.50 \pm 212.56$ |
| 20 | 20 | $348.50 \pm 144.20$ | $141.30 \pm 70.23$ | $500.00 \pm 0.00$ |
| 50 | 20 | $325.80 \pm 139.27$ | $146.60 \pm 36.67$ | $500.00 \pm 0.00$ |
| 100 | 20 | $324.60 \pm 124.09$ | $145.30 \pm 48.06$ | $450.00 \pm 130.99$ |
| 5 | 50 | $339.80 \pm 139.34$ | $174.60 \pm 11.71$ | $500.00 \pm 0.00$ |
| 10 | 50 | $346.70 \pm 128.98$ | $174.50 \pm 12.89$ | $500.00 \pm 0.00$ |
| 20 | 50 | $341.80 \pm 123.77$ | $172.40 \pm 19.69$ | $500.00 \pm 0.00$ |
| 50 | 50 | $330.10 \pm 120.47$ | $177.80 \pm 10.39$ | $500.00 \pm 0.00$ |
| 100 | 50 | $326.00 \pm 151.14$ | $178.10 \pm 17.01$ | $500.00 \pm 0.00$ |
| 5 | 100 | $337.70 \pm 115.30$ | $135.90 \pm 76.44$ | $500.00 \pm 0.00$ |
| 10 | 100 | $329.00 \pm 122.08$ | $148.40 \pm 104.72$ | $500.00 \pm 0.00$ |
| 20 | 100 | $328.90 \pm 124.00$ | $124.10 \pm 47.61$ | $500.00 \pm 0.00$ |
| 50 | 100 | $332.20 \pm 123.10$ | $129.50 \pm 61.10$ | $500.00 \pm 0.00$ |
| 100 | 100 | $386.40 \pm 132.20$ | $136.30 \pm 93.81$ | $500.00 \pm 0.00$ |
| 5 | 200 | $354.00 \pm 126.69$ | $158.40 \pm 53.20$ | $500.00 \pm 0.00$ |
| 10 | 200 | $331.90 \pm 130.53$ | $169.90 \pm 93.20$ | $500.00 \pm 0.00$ |
| 20 | 200 | $371.20 \pm 133.87$ | $176.60 \pm 81.36$ | $500.00 \pm 0.00$ |
| 50 | 200 | $332.80 \pm 125.28$ | $175.60 \pm 62.60$ | $500.00 \pm 0.00$ |
| 100 | 200 | $340.60 \pm 144.32$ | $199.20 \pm 97.88$ | $500.00 \pm 0.00$ |

## H.5 COMPARISION AGAINST BEHAVIOR CLONING DISTILLATION

Behavior cloning (BC) based approaches (like those of Lei et al. (2024), Light et al. (2024)) do not work well on datasets generated by a non-expert policy. On the other hand, the Cartpole dataset used in our paper was generated by a completely random policy. Nevertheless, we propose to demonstrate this empirically by adding the following experiments comparing the performance of our method with the BC loss based distillation from (Light et al., 2024). We observe that BC on the full training data as well as the synthetic dataset created using the algorithm from (Light et al., 2024) perform much worse than our techniques as the former do not take rewards into consideration. This demonstrates the dependence on data quality of BC based distillation approaches and the effectiveness of our method.

Evaluation on the Cartpole Random Dataset of $10K$ transitions, comparing against BC and BC on synthetic data generated by (Light et al., 2024) with $k = 100$ sampled models is shown in Table 5.

Table 5: Cartpole Random Dataset, comparing against BC on full training data and $BC_{syn}$ (Light et al., 2024)

| $N_{syn}$ | $D_{train}$ | $D_{rand}$ | $D_{syn}$ | $BC$ | $BC_{syn}$ |
|---|---|---|---|---|---|
| 10 | $368.20 \pm 116.77$ | $13.30 \pm 1.19$ | $464.00 \pm 108.00$ | $17.50 \pm 5.46$ | $13.30 \pm 1.68$ |
| 20 | $324.60 \pm 124.09$ | $145.30 \pm 48.06$ | $450.00 \pm 130.99$ | $19.30 \pm 9.17$ | $13.50 \pm 5.14$ |
| 50 | $326.00 \pm 151.14$ | $178.10 \pm 17.01$ | $500.00 \pm 0.00$ | $28.30 \pm 14.95$ | $17.10 \pm 12.86$ |
| 100 | $386.40 \pm 132.20$ | $136.30 \pm 93.81$ | $500.00 \pm 0.00$ | $26.00 \pm 13.51$ | $14.20 \pm 4.02$ |
| 200 | $340.60 \pm 144.32$ | $199.20 \pm 97.88$ | $500.00 \pm 0.00$ | $17.60 \pm 6.87$ | $22.90 \pm 8.46$ |

## H.6 Additional Experiments and Details for the Mountain Car Environment

The Mountain Car MDP (Moore (1990), Towers et al. (2024) MIT License) is a deterministic MDP that consists of a car placed stochastically at the bottom of a sinusoidal valley, with the only possible actions being the accelerations that can be applied to the car in either direction. The observation space is continuous with two dimensions and there are three discrete actions possible. The goal is to reach the flag placed on top of the right hill as quickly as possible, as such the agent is penalised with a reward of $-1$ for each timestep. The episode truncates after 200 timesteps and is also terminated if the position of the car goes beyond a certain range.

To generate the offline data, we sample 5000 samples through a uniformly random policy and another 5000 samples through an expert policy trained using tabular $q$-learning on a discretized state space.

We use the same learning rate search method as mentioned in Appendix H.4 and optimizers to get the synthetic data, partitioned into terminated and non-terminated samples. We use a neural network with two hidden layers with 64 neurons and ReLU activations for the $q$-value predictor with the Fitted-Q iteration algorithm for training. Policies trained on the three datasets are evaluated in the environment and the average returns over 10 trials are reported in Table 6. We observe that the policies trained on the random datasets are not able to reach the top of the hill and are all truncated. Policies trained on the synthetic data are on average able to reach the top of the hill, albeit slower than ones trained on the full offline dataset. As in the Cartpole environment, increasing $N_{\text{syn}}$ increases our performance, but increasing $k$ has no impact on performance.

Table 6: Evaluation for Mountain Car with $(64, 64)$-layer NN.

| $k$ | $N_{\text{syn}}$ | $D_{\text{train}}$ | $D_{\text{rand}}$ | $D_{\text{syn}}$ |
|---|---|---|---|---|
| 5 | 10 | -111.40 ± 13.16 | -200.00 ± 0.00 | -200.00 ± 0.00 |
| 10 | 10 | -109.80 ± 27.92 | -200.00 ± 0.00 | -200.00 ± 0.00 |
| 20 | 10 | -111.60 ± 15.10 | -200.00 ± 0.00 | -157.70 ± 2.28 |
| 50 | 10 | -110.10 ± 22.39 | -200.00 ± 0.00 | -200.00 ± 0.00 |
| 100 | 10 | -106.50 ± 15.23 | -200.00 ± 0.00 | -195.90 ± 8.28 |
| 5 | 50 | -106.30 ± 15.49 | -200.00 ± 0.00 | -183.90 ± 24.61 |
| 10 | 50 | -104.80 ± 15.14 | -200.00 ± 0.00 | -200.00 ± 0.00 |
| 20 | 50 | -104.50 ± 13.97 | -200.00 ± 0.00 | -129.50 ± 30.31 |
| 50 | 50 | -108.90 ± 16.26 | -200.00 ± 0.00 | -162.60 ± 17.24 |
| 100 | 50 | -106.90 ± 14.45 | -200.00 ± 0.00 | -159.00 ± 25.60 |
| 5 | 200 | -109.80 ± 16.02 | -200.00 ± 0.00 | -149.60 ± 25.39 |
| 10 | 200 | -108.50 ± 18.70 | -200.00 ± 0.00 | -133.50 ± 3.14 |
| 20 | 200 | -105.40 ± 14.71 | -200.00 ± 0.00 | -121.90 ± 37.87 |
| 50 | 200 | -112.60 ± 14.99 | -200.00 ± 0.00 | -132.00 ± 23.00 |
| 100 | 200 | -105.30 ± 14.16 | -200.00 ± 0.00 | -162.90 ± 32.98 |

## H.7 Additional Experiments and Details for the Acrobot Environment

The Acrobot Environment ((Sutton, 1995), (Towers et al., 2024) MIT License) consists of two links connected linearly to form a chain, with one end of the chain fixed. The joint between the two links is actuated. The goal is to apply torques on the actuated joint to swing the free end of the linear chain above a given height while starting from the initial state of hanging downwards. The action space is discrete with three possible actions and the state space is continuous with 6 dimensions. The goal is to have the free end reach a designated target height in as few steps as possible, and as such all steps that do not reach the goal incur a reward of $-1$. Achieving the target height results in termination with a reward of 0. Episodes are artificially truncated after 500 timesteps.

To generate the offline data, we sample 10000 transitions through a uniformly random policy. We use the same learning rate search method and optimizers as mentioned in Appendix H.4 to get the synthetic data, partitioned into terminated and non-terminated samples. We use a neural network with two hidden layers with 64 neurons and ReLU activations for the $q$-value predictor with the Fitted-Q iteration algorithm for training. Policies trained on the three datasets are evaluated in the environment and the average returns over 10 trials are reported in Table 7. We observe that policies trained on the synthetic data give higher returns on average than policies trained on the full offline dataset and a

randomly subsampled smaller dataset. As in the Cartpole environment, increasing $N_{\text{syn}}$ increases our performance, but increasing $k$ as no impact on performance.

Table 7: Evaluation for Acrobot with $(64, 64)$-layer NN.

| $k$ | $N_{\text{syn}}$ | $D_{\text{train}}$ | $D_{\text{rand}}$ | $D_{\text{syn}}$ |
|---|---|---|---|---|
| 5 | 10 | $-84.20 \pm 10.82$ | $-500.00 \pm 0.00$ | $-74.10 \pm 7.49$ |
| 10 | 10 | $-82.30 \pm 12.04$ | $-500.00 \pm 0.00$ | $-74.60 \pm 9.65$ |
| 20 | 10 | $-81.30 \pm 4.86$ | $-500.00 \pm 0.00$ | $-77.50 \pm 5.39$ |
| 50 | 10 | $-84.60 \pm 11.88$ | $-500.00 \pm 0.00$ | $-72.70 \pm 6.99$ |
| 100 | 10 | $-84.20 \pm 16.61$ | $-500.00 \pm 0.00$ | $-74.40 \pm 9.97$ |
| 5 | 50 | $-85.10 \pm 10.97$ | $-479.20 \pm 41.97$ | $-75.00 \pm 10.92$ |
| 10 | 50 | $-83.40 \pm 11.24$ | $-481.30 \pm 38.43$ | $-77.40 \pm 7.42$ |
| 20 | 50 | $-82.60 \pm 8.85$ | $-481.50 \pm 37.49$ | $-75.80 \pm 9.40$ |
| 50 | 50 | $-85.50 \pm 9.45$ | $-480.90 \pm 38.23$ | $-78.00 \pm 8.80$ |
| 100 | 50 | $-83.30 \pm 13.24$ | $-484.70 \pm 30.60$ | $-75.00 \pm 10.24$ |
| 5 | 200 | $-85.60 \pm 7.23$ | $-103.30 \pm 13.99$ | $-89.10 \pm 6.61$ |
| 10 | 200 | $-83.10 \pm 11.07$ | $-99.00 \pm 10.74$ | $-86.90 \pm 20.87$ |
| 20 | 200 | $-82.40 \pm 8.97$ | $-99.20 \pm 8.40$ | $-88.20 \pm 9.94$ |
| 50 | 200 | $-86.90 \pm 5.26$ | $-102.50 \pm 4.20$ | $-85.70 \pm 11.71$ |
| 100 | 200 | $-85.10 \pm 8.58$ | $-102.30 \pm 9.96$ | $-84.50 \pm 13.40$ |

