# OpenReview forum: "Algorithmic Guarantees for Distilling Supervised and Offline RL Datasets"
_ICLR.cc/2026/Conference — ICLR 2026 Poster_

### Official Review · Reviewer_4Kx3 · 2025-10-31

**Soundness:** 3
**Presentation:** 3
**Contribution:** 3
**Rating:** 6
**Confidence:** 1

**Summary:**

This paper proposes a dataset distillation algorithm by matching the losses between training and synthetic datasets without introducing extra model training. The authors develop and analyze a method for supervised learning and offline reinforcement learning, where only $\hat{\mathcal{O}}(d^2)$ regressors are sufficient to ensure the MSE loss of any bounded linear model is approximately preserved. They also provide a matching lower bound of  $\Omega (d^2)$, establishing tightness. Furthermore, the authors extend the algorithm beyond supervised learning, i.e., offline RL, by leveraging the next state and reward information in the dataset to match the Bellman loss rather than the behavior cloning loss. Extensive theoretical analysis and supplementary experiments prove that the proposed method efficiently distills the synthetic dataset from the training dataset without relying on auxiliary techniques-additional classifier training.

**Strengths:**

- The paper is well structured and addresses comprehensive details.
- Extensive theoretical proofs concretely support the main claim.
- Experiments demonstrate that the proposed method shows a clear margin compared to other baselines, recovering near-optimal or even outperforming performance.

**Weaknesses:**

- The experimental section appears relatively narrow, focusing on small regression settings with Gym control tasks. Comparing with other baselines (Lei et al. 2024, Light et al. 2024) for generating synthetic datasets would improve the soundness of the suggested method.

**Questions:**

- What would be the major bottleneck when extending the proposed method to a non-linear function approximator (i.e., neural network) with offline RL?

---

> ### Author Response · Authors · 2025-11-23
> **Author Response to Reviewer 4Kx3**
>
> We thank the Reviewer for their helpful feedback and we address their concerns below.
>
>
> *R: The experimental section appears relatively narrow, focusing on small regression settings with Gym control tasks. Comparing with other baselines (Lei et al. 2024, Light et al. 2024) for generating synthetic datasets would improve the soundness of the suggested method.*
>
> A: Behavior cloning (BC) based approaches (like those from Lei et al. 2024, Light et al. 2024)  do not work well on datasets generated by a non-expert policy. On the other hand, the Cartpole dataset used in our paper was generated by a completely random policy.
> Nevertheless, we propose to demonstrate this empirically by adding the following experiments comparing the performance of our method with the BC loss based distillation from Light et al. We observe that BC on the full training data as well as the synthetic dataset created using the algorithm from Light et al. perform much worse than our techniques as the former do not take rewards into consideration. This demonstrates the dependence on data quality of BC based distillation approaches and the effectiveness of our method.
>
> Cartpole Random Dataset of 10k transitions, comparing against BC and (Light et al.) with
> k=100 (sampled models)
>
> | $N_{\textnormal{syn}}$|  $D_{\textnormal{train}}$ | $D_{\textnormal{rand}}$ | $D_{\textnormal{syn}}$ | $BC$ |$BC_{\textnormal{syn}}$  (Light et al.) |
> | :--- | :---: | :---: | :---: | :---: | :---: |
> |  $10$ | $368.20 \pm 116.77$ | $13.30 \pm 1.19$ | $464.00 \pm 108.00$ | $17.50 \pm 5.46$ | $13.30 \pm 1.68$|
> |  $20$ | $324.60 \pm 124.09$ | $145.30 \pm 48.06$ | $450.00 \pm 130.99$ | $19.30 \pm 9.17$ | $13.50 \pm 5.14$|
> |  $50$ | $326.00 \pm 151.14$ | $178.10 \pm 17.01$ | $500.00 \pm 0.00$ | $28.30 \pm 14.95$ | $17.10 \pm 12.86$|
> |  $100$ | $386.40 \pm 132.20$ | $136.30 \pm 93.81$ | $500.00 \pm 0.00$ |$26.00 \pm 13.51$| $14.20 \pm 4.02$|
> |  $200$ | $340.60 \pm 144.32$ | $199.20 \pm 97.88$ | $500.00 \pm 0.00$ | $17.60 \pm 6.87$ |$22.90 \pm 8.46$ |
>
> $ $
>
> *R: What would be the major bottleneck when extending the proposed method to a non-linear function approximator (i.e., neural network) with offline RL?*
> A: For the offline RL case we use a commonly used assumption of a feature-map and linear action-value predictors. Our analysis uses a conditioning argument which leverages the linearity of the predictor for using Gaussian anti-concentration. The anti-concentration is used to show that the sampled linear predictors lie in an appropriate subset with non-trivial probability. By assuming or proving appropriate Lipschitzness of a non-linear predictor’s output w.r.t. to its weights, we believe our arguments can be extended to such predictors, e.g. neural networks. We however, defer a more detailed study of the non-linear predictor case for the future. Nevertheless, we shall include these insights in the final version of the paper.

---

> > ### Comment · Reviewer_4Kx3 · 2025-11-25
> >
> > Thank you for the comprehensive explanations. Further comparisons across different sizes of $N_\text{syn}$ with BC confirm the effectiveness of the suggested distillation method. My concerns about weaknesses and questions are mostly resolved.

---

### Official Review · Reviewer_Ham7 · 2025-11-01

**Soundness:** 3
**Presentation:** 3
**Contribution:** 3
**Rating:** 6
**Confidence:** 2

**Summary:**

This paper introduces a dataset distillation method for supervised regression and offline RL that matches the loss on the original and synthetic datasets with respect to a fixed set of randomly sampled models, avoiding bi-level optimization. The key contribution is a theoretical guarantee that only $O(d^{²})$ sampled linear regressors are needed for the supervised case, with a matching lower bound. The experiments demonstrate its effectiveness.

**Strengths:**

1. The paper provides theoretical guarantees for dataset distillation, an area where such analysis is often lacking. The upper and matching lower bounds for the supervised case are particularly compelling.
2. The experiments adequately support the theoretical claims, showing that the method works well in practice, even with non-linear neural networks, and outperforms baseline approaches like random subsampling.

**Weaknesses:**

1. The empirical evaluation is limited to relatively small-scale datasets and standard RL benchmarks. A more extensive evaluation on larger-scale or more complex datasets would strengthen the claims of practical efficacy.

**Questions:**

1. Could the authors discuss potential pathways for extending the theoretical guarantees to non-linear function approximators, such as neural networks? Note that I'm not asking for additional experimental results, just want to have a discussion, since making a more constructive theoretical analysis on non-linear functions is more feasible and practical for most real-world cases.
2. For the offline RL setting, how restrictive is the decomposable feature map assumption in practice?

---

> ### Author Response · Authors · 2025-11-23
> **Author Response to Reviewer Ham7**
>
> We thank the Reviewer for their helpful feedback and we address their concerns below.
>
>
> *R: The empirical evaluation is limited to relatively small-scale datasets and standard RL benchmarks. A more extensive evaluation on larger-scale or more complex datasets would strengthen the claims of practical efficacy.*
>
> A:  For the supervised case we propose to add experiments on the California Housing dataset [1] which has $\sim 20K$ points and $9$ features. The dataset distillation is done using linear models, while the training and evaluation is done using a $(10,10)$ two hidden layer ReLU based neural network, demonstrating that our synthetic data generation process can in practice be used with different model architectures for evaluation. We sample $k=100$ random linear models for distillation.
>
>
> $ $
>
> California Housing (20640 samples, 9 features)
>
> | $N\_{\textnormal{syn}}$|  $D\_{\textnormal{train}}$ | $D\_{\textnormal{rand}}$ | $D\_{\textnormal{syn}}$ |
> | :--- | :---: | :---: | :---: |
> |  $10$ | $0.56\pm 0.05$ | $1.11\pm 0.27$ | $0.99\pm 0.05$ |
> |  $20$ | $0.56 \pm 0.05$ | $1.13\pm 0.33$ | $0.94\pm 0.07$ |
> |  $50$ |  $0.56 \pm 0.05$ | $1.16\pm 0.30$ | $0.73\pm 0.06$ |
> |  $100$ | $0.56 \pm 0.05$ | $1.16\pm 0.71$ | $0.73\pm 0.07$ |
> |  $200$ | $0.56 \pm 0.05$ | $0.94\pm 0.38$ | $0.69\pm 0.02$ |
>
> We see from the above that our distillation technique significantly outperforms random sampling and achieves $\sim 200$-factor compression, even on large datasets like California Housing.
>
> *RL benchmarks*: Modern large scale Offline RL benchmarks like D4RL are created to distinguish between standard offline RL algorithms like FQI (which does iterative Bellman Loss minimization) and recent advanced algorithms like CQL (see pg. 2 of [2]). While our paper proposes a novel offline RL dataset distillation algorithm based on Bellman Loss matching, since D4RL is unsuited for Bellman Loss based algorithms, one would need to adapt dataset distillation to more complicated loss functions like that used in CQL. These generalizations of our techniques, their analysis and evaluations on D4RL benchmarks are beyond the scope of this paper and are deferred to future work.
>
> $ $
>
> *R: Could the authors discuss potential pathways for extending the theoretical guarantees to non-linear function approximators, such as neural networks?*
> A: Our theoretical analysis of the supervised regression setting can be carried out for any class of non-linear predictors that admit anti-concentration of $L^{\textnormal{mse}}\_{\textnormal{err}}$ (eqn. (1)) over the choice of the predictor's random weights. In the linear setting, $L^{\textnormal{mse}}\_{\textnormal{err}}$ is a degree 4 polynomial in the weights of the linear predictor and we use the Carbery-Wright inequality for polynomials (Theorem A.3 in Appendix A.3) to get anti-concentration in Lemma C.1. If some class of neural networks can be approximated by polynomials of bounded degree (as suggested by [3] below) whose coefficients can be sampled randomly, our techniques could be applied. However, proving such anti-concentration for general neural networks is beyond the scope of this paper and we defer this line of work to the future. For the offline RL case we use a commonly used assumption of a feature-map and linear action-value predictors. Our analysis uses a conditioning argument which leverages the linearity of the predictor for using Gaussian anti-concentration. The anti-concentration is used to show that the sampled linear predictors lie in an appropriate subset with non-trivial probability. By assuming or proving appropriate Lipschitzness of a non-linear predictor’s output w.r.t. to its weights, we believe our arguments can be extended to such predictors, e.g. neural networks. We however, defer a more detailed study of the non-linear predictor case for the future. Nevertheless, we shall include these insights in the final version of the paper.
>
> $ $
>
> **response continued below..**

---

> > ### Author Response · Authors · 2025-11-23
> >
> > *R: For the offline RL setting, how restrictive is the decomposable feature map assumption in practice?*
> > A: As mentioned on lines 350-354, decomposable maps are commonly used in value based deep RL methods. Typically, the feature maps of the state and action are concatenated and a neural network is used to predict the $Q$-value of the concatenated representation. In Theorem 4.4 we prove an efficient dataset distillation bound for the case of linear $Q$-value predictors, and we believe that extending this to commonly used neural networks is a challenging and interesting direction for future work.
> >
> > [1] Pace, R. K., & Barry, R. (1997). Sparse spatial autoregressions. Statistics & Probability Letters, 33(3), 291–297. https://doi.org/10.1016/S0167-7152(96)00140-X
> >
> > [2] Fu, J., Kumar, A., Nachum, O., Tucker, G., and Levine, S. (2020). D4rl: Datasets for deep data-driven reinforcement learning. In arXiv. Link: https://arxiv.org/pdf/2004.07219
> >
> > [3] Zhu Frances , Jing Dongheng , Leve Frederick , Ferrari Silvia. “NN-Poly: Approximating common neural networks with Taylor polynomials to imbue dynamical system constraints”, Frontiers in Robotics and AI, volume 9, 2022.

---

### Official Review · Reviewer_3Mno · 2025-11-08

**Soundness:** 3
**Presentation:** 3
**Contribution:** 3
**Rating:** 4
**Confidence:** 3

**Summary:**

The paper tackles supervised regression and offline RL settings by constructing synthetic datasets that preserve the original objective of training with original training dataset. It introduces a loss-matching method using randomly sampled linear regressors/Q-value predictors achieving Õ(d^2) sampling guarantees for regression (alongside a matching Ω(d^2) lower bound) and exp(O(d log d)) in offline RL (furthermore relaxed Õ(d^2) under decomposable feature maps). The approach shows solid empirical performance on standard regression datasets and classic offline RL benchmarks, and offers a timely theory-first approach for distillation.

**Strengths:**

- The use of loss-matching method using randomly sampled linear regressors/Q-value predictors is particularly important tool that is used in RL literature.
- Two results for each setting look comprehensive as contributions to the learning community.
- Experiments demonstrate that small synthetic sets and few sampled models can perform competitively on standard regression datasets and classic offline RL benchmarks.

**Weaknesses:**

- Even though experiments demonstrate that small synthetic sets match the performance when trained with entire training dataset, some issues pop up:
- - This phenomena cannot be explained by the current theory. The distillation problem is only interesting for the case $n>>m$. This is also the focus of the literature (Light et al. 25, Lei et al. 24) cited by this work.
- - Lemma C.4 does provide net arguments for the choice of synthetic dataset size. But it seems that both $m$ and $n$ are proportional to $d$, whereas the dependence on $\epsilon$ for $m$ is unclear.
- - The training size for toy sequential decision problems are high. I suggest authors to really use established benchmarks like D4RL to demonstrate their distillation behavior.

**Questions:**

- na -

---

> ### Author Response · Authors · 2025-11-23
> **Author Response to Reviewer 3Mno**
>
> We thank the Reviewer for their helpful feedback and we address their concerns below.
>
> *R: This phenomena cannot be explained by the current theory. The distillation problem is only interesting for the case $n \gg m$. This is also the focus of the literature (Light et al. 25, Lei et al. 24) cited by this work.*
> A: We wish to respectfully emphasize that our work **does handle** the case of $n \gg m$. For the supervised case, Theorem 4.1 shows that for a given $n$-sized training dataset $D^{\textsf{sup}}\_{\textnormal{train}}$, with high probability over the sampled regressors, if there exists *any* $m$-sized synthetic dataset $D^{\textsf{sup}}\_{\textnormal{syn}}$ which satisfies (5) i.e., nearly matches the losses on the sampled regressors with $D^{\textsf{sup}}\_{\textnormal{train}}$, then it is a good distilled synthetic dataset. There is no restriction on $m$, as long as (5) is satisfied. Similarly, in the offline RL case, there is no restriction on $m = |D^{\textsf{orl}}\_{\textnormal{syn}}|$ as long as the condition of Theorem 4.3 (or Theorem 4.4) is satisfied.
> We would like to point out that a small value of $m$ (independent of $n$) is always possible, as evidenced by Lemma D.2 in the supervised case which shows that if the norm bounds for $D^{\textsf{sup}}\_{\textnormal{syn}}$ are relaxed then such a $D^{\textsf{sup}}\_{\textnormal{syn}}$ of size $m = q+1$ always exists. This argument also holds for the offline RL case of linearly decomposable maps. Similar upper bounds can also be obtained using pseudo-dimension based generalization error bounds (see for e.g.  Theorem 17.1 of [Anthony-Bartlett, 2009]), which are also applicable to the general offline RL case (as in Lemma E.4). We will include a summary of this explanation in the paper.
>
> *R: Lemma C.4 does provide net arguments for the choice of synthetic dataset size. But it seems that both $m$ and $n$ are proportional to $d$, whereas the dependence on $\varepsilon$ for $m$ is unclear.*
> A: In Lemma C.4 we replace the $m$-sized synthetic dataset $D^{\textsf{sup}}\_{\textnormal{syn}}$ with a dataset $\hat{D}$ of size $q = d+1$ with points from an appropriate net. This allows us to take a union bound over all possible $q$-sized $\hat{D}$. The value of $m$ is unrestricted, and need not depend on $d$. There is no restriction on $n = |D^{\textsf{sup}}\_{\textnormal{train}}|$ either, and we anyway do not require a net argument for $D^{\textsf{sup}}\_{\textnormal{train}}$.
> Due to this net argument which replaces  $D^{\textsf{sup}}\_{\textnormal{syn}}$  with $q$-sized $\hat{D}$ (albeit with possibly larger norms), the analysis is made independent of $m$ and thus $\varepsilon$ and $\delta$ are also independent of $m$.
>
> *R: The training size for toy sequential decision problems are high. I suggest authors to really use established benchmarks like D4RL to demonstrate their distillation behavior.*
> A: The D4RL benchmark is created to distinguish between standard offline RL algorithms like FQI (which does iterative Bellman Loss minimization) and recent advanced algorithms like CQL (see pg. 2 of [1]). While our paper proposes a novel offline RL dataset distillation algorithm based on Bellman Loss matching, since D4RL is unsuited for Bellman Loss based algorithms, one would need to adapt dataset distillation to more complicated loss functions like that used in CQL. These generalizations of our techniques, their analysis and evaluations on D4RL benchmarks are beyond the scope of this paper and are deferred to future work.
>
>
> [1] Fu, J., Kumar, A., Nachum, O., Tucker, G., and Levine, S. (2020). D4rl: Datasets for deep data-driven reinforcement learning. In arXiv. Link: https://arxiv.org/pdf/2004.07219

---

> > ### Comment · Reviewer_3Mno · 2025-11-26
> >
> > Thanks for the clarifications. I encourage authors to include these discussions in the main paper after Thm 4.2 or 4.4. Adding such examples to support the regime $n \gg m$ helps future research.
> >
> > Regarding Lemma C.4: *The value of $m$ is unrestricted, and need not depend on $d$.* Why? I am more confused since $m$-sized set construction depends on $q=d+1$-net sets. Also, can we actually characterize hardness of such construction? i.e. lower bound for $m$. The unrestricted part gives way to constant sized regimes, which is hard to intuitively satisfy if one has growing $n$.
> >
> > Please do include this discussion on high-samples for toy sequential decision problems and extensions to D4RL or other benchmarks as *limitations*.

---

> > > ### Author Response · Authors · 2025-11-27
> > >
> > > We thank the Reviewer for their reply and address their concern regarding the value of $m$ as follows:
> > > We wish to clarify that Theorem 4.1 states that as long as there exists $D^{\textsf{sup}}\_{\textnormal{syn}}$ of size $m$ such that (5) is satisfied, the implications of the theorem hold. In particular, Theorem 4.1 does not explicitly impose any lower bound on $m$. However, depending on  $D^{\textsf{sup}}\_{\textnormal{train}}$, it may not always be possible to satisfy (5) for small values of $m$. Specifically, we show in Appendix D.1 a lower bound on $m$ based on the spectral properties of the gram matrix of $D^{\textsf{sup}}\_{\textnormal{train}}$. In the worst case, we show a lower bound of $m \geq d+1$, as also mentioned on line 325.
> > > $ $
> > >
> > > As suggested by the Reviewer, we will include discussions on the size of the synthetic dataset, and examples for the $n \gg m$ regime after the statements of the relevant theorems. Also, we will mention the challenges in applying our techniques to the D4RL benchmark as limitations.

---

> > > > ### Comment · Reviewer_3Mno · 2025-11-27
> > > >
> > > > Thank you for the clarifications. I am increasing my score from 4 to 6 as my concerns are mostly resolved with the assumption that the revision will contain such discussions for future readers.

---

### Meta-Review · Area_Chair_4TXg · 2026-01-09

**Summary:**

The reviewers generally lauded the paper for its novel theoretical contributions in dataset distillation for both supervised learning and offline reinforcement learning, particularly highlighting the algorithmic guarantees and the matching lower bounds for the supervised case. They appreciated the theory-first approach and the empirical validation provided. However, several common concerns emerged, primarily revolving around the scope of the empirical evaluation, the applicability and extension of theoretical guarantees to more complex, non-linear models like neural networks, and specific clarifications regarding the theoretical framework, such as the n >> m regime and the role of synthetic dataset size. Reviewers also sought comparisons with existing behavioral cloning-based distillation methods and a discussion on the limitations of the current approach, especially concerning large-scale offline RL benchmarks like D4RL.

**Reviewer Concerns:**

Many of the initial concerns raised by the reviewers were effectively addressed during the rebuttal period. Reviewers 3Mno, Ham7, and 4Kx3 all questioned the limited empirical evaluation, suggesting larger datasets and comparisons with other baselines. The authors responded by adding experiments on the California Housing dataset for supervised learning and a comparison with Light et al.'s behavioral cloning method on Cartpole for offline RL, which largely satisfied these concerns. Reviewer 3Mno's query about the n >> m regime and the synthetic dataset size (m) in Lemma C.4 was clarified by the authors, explaining that their theory does cover this case and m is not restricted in the way initially perceived, leading to the reviewer increasing their score. The general question regarding extending theoretical guarantees to non-linear function approximators (neural networks) was addressed by the authors through detailed discussions on potential pathways involving anti-concentration and Lipschitzness, though acknowledging these are challenging directions for future work. Similarly, the restrictiveness of the decomposable feature map assumption in offline RL and the reasons for not using D4RL benchmarks were explained, with the authors agreeing to include these discussions as limitations in the final paper. While the discussions on non-linear extensions provided valuable insight, a full theoretical solution for neural networks remains an outstanding challenge, explicitly deferred to future work, which is a common and reasonable outcome for such complex theoretical problems.

**Reviewer Scores:**

Reviewer 3Mno explicitly increased their score from 4 ("marginally below the acceptance threshold") to 6 ("marginally above the acceptance threshold") after the rebuttal, indicating that their concerns were mostly resolved. Reviewer Ham7 initially gave a score of 6 ("marginally above the acceptance threshold") with confidence 2, and their primary concerns regarding empirical scope and theoretical extensions were addressed with additional experiments and discussions. It is highly probable that Reviewer Ham7 would maintain their score of 6, potentially with a slightly increased confidence, given the authors' comprehensive responses. Reviewer 4Kx3, despite initially having very low confidence (1) and flagging the paper as out-of-expertise, still rated it 6 ("marginally above the acceptance threshold"). Following the rebuttal, they confirmed their concerns were "mostly resolved," especially after seeing the additional comparisons. Therefore, Reviewer 4Kx3 would also likely maintain their score of 6, and given the resolution of their experimental and conceptual questions, their confidence in the assessment would likely improve.

---

### Decision · Program_Chairs · 2026-01-26

Accept (Poster)